# Analysis of Circulating Tumor and Cancer Stem Cells Provides New Opportunities in Diagnosis and Treatment of Small Cell Lung Cancer

**DOI:** 10.3390/ijms231810853

**Published:** 2022-09-17

**Authors:** Evgenii G. Skurikhin, Natalia Ermakova, Mariia Zhukova, Olga Pershina, Edgar Pan, Angelina Pakhomova, Lena Kogai, Victor Goldberg, Elena Simolina, Victoria Skurikhina, Darius Widera, Aslan Kubatiev, Sergey G. Morozov, Nikolai Kushlinskii, Alexander Dygai

**Affiliations:** 1Laboratory of Regenerative Pharmacology, Goldberg ED Research Institute of Pharmacology and Regenerative Medicine, Tomsk National Research Medical Centre of the Russian Academy of Sciences, Lenin, 3, 634028 Tomsk, Russia; 2Ministry of Health of the Russian Federation, Siberian State Medical University, Moskovski, 2, 634050 Tomsk, Russia; 3Cancer Research Institute, Tomsk National Research Medical Center, Kooperativny, 5, 634009 Tomsk, Russia; 4Stem Cell Biology and Regenerative Medicine Group, School of Pharmacy, Whiteknights Campus, Reading RG6 6AP, UK; 5Institute of General Pathology and Pathophysiology, 125315 Moscow, Russia; 6Blokhin National Medical Research Center of Oncology, 115522 Moscow, Russia

**Keywords:** small cell lung cancer, cancer stem cells, circulating tumor cells, reprogrammed CD8^+^ T-lymphocytes

## Abstract

Current methods for diagnosis and treatment of small cell lung cancer (SCLC) have only a modest efficacy. In this pilot study, we analyzed circulating tumor cells (CTCs) and cancer stem cells (CSCs) in patients with SCLC to search for new diagnostic and prognostic markers and novel approaches to improve the treatment of the disease. In other forms of lung cancer, we showed a heterogeneity of blood CTCs and CSCs populations, as well as changes in other cell populations (ALDH^+^, CD87^+^CD276^+^, and EGF^+^Axl^+^) in smokers. A number of CTCs and CSCs in patients with SCLC have been shown to be resistant to chemotherapy (CT). High cytotoxic activity and resistance to apoptosis of reprogrammed CD3^+^CD8^+^ T-lymphocytes (rTcells) in relation to naive CD3^+^CD8^+^ T-lymphocytes was demonstrated in a smoking patient with SCLC (Patient G) in vitro. The target for rTcells was patient G’s blood CSCs. Reprogramming of CD3^+^CD8^+^ T-lymphocytes was carried out with the MEK1/2 inhibitor and PD-1/PD-L1 pathway blocker nivolumab. The training procedure was performed with a suspension of dead CTCs and CSCs obtained from patient’s G blood. The presented data show a new avenue for personalized SCLC diagnosis and targeted improvement of chemotherapy based on the use of both CTCs and CSCs.

## 1. Introduction

Lung cancer is one of the most common forms of cancer with its incidence increasing every year. In 2018, there were 2.09 million new cases of lung cancer and 1.76 million associated deaths [1]. In 2020, these figures increased to 2.21 million and 1.80 million, respectively [2]. According to recent forecasts, more than 230,000 new cases of lung and bronchial cancer and about 130,000 deaths will be registered in the USA alone in 2022 [3]. SCLC accounts for about 15% of all lung cancer cases. SCLC is the most aggressive form of lung cancer. It is characterized by asymptomatic course, high rate of progression and metastasis, and poor prognosis. The 5-year survival rates for SCLC vary depending on the stage, but the average is about 7% [4].

Chemotherapy and radiation therapy remain an important component of SCLC treatment [5,6]. However, these SCLC treatments are not very effective [7]. In addition, despite the high sensitivity to first-line chemotherapy, SCLC tends to be resistant to further courses of chemotherapy [8]. Immunotherapy is a promising approach for the treatment of SCLC. Indeed, the use of immune checkpoint inhibitors improves the quality of life of patients with SCLC and increases their life expectancy [9]. However, the effectiveness of immunotherapy in SCLC is still modest [10].

Importantly, the issue of early diagnosis of SCLC remains unresolved. Despite the existing range of diagnostic methods, including radiation diagnostics, cytological, and histological methods, the disease is often diagnosed late, with 70% of patients already in stage IV at the time of diagnosis when the clinical symptoms are pronounced and the disease progresses. Treatment at this stage is palliative [4,11]. Lack of methods for detecting the disease at its early stages delays the treatment, negatively affecting the outcome of the disease. In addition, tumor resistance to chemotherapeutic agents makes it important to regularly monitor the effectiveness of therapy.

Therapy resistance, cancer metastasis, and disease recurrence are associated with cancer stem cells (CSCs) [5], which are one of the types of circulating tumor cells (CTCs). Due to the rapid spread of the tumor and the large number of CTCs in SCLC, a relationship between CTCs and CSCs is suggested [12]. Despite the fact that CSCs from different tumors have been phenotypically and functionally characterized, there are still no specific markers that would clearly define CSC. CD44, CD87, CD117, CD276, Axl, EGF, and Ki67 are the most commonly used markers to identify CSCs in lung cancer [13,14]. In a previous study, we showed that the determination of circulating CSCs in patients with breast cancer has diagnostic and therapeutic value [15]. CTCs found in patients with cancer circulate either as single cells or as small clusters of CTCs called CTCs spheroids [16]. It has been shown that they have an increased metastatic potential and ability to spontaneously form large spheroids called tumorspheres in vitro. It is believed that CSCs have the potential to form spheroids. In particular, the formation of neurospheres and mammospheres by various types of cancer cells has been demonstrated [17].

The immune system determines the nature and effectiveness of the antitumor response. CD8^+^ T-lymphocytes, secreting perforins and granzymes, exhibit cytotoxic activity against tumor cells [18]. However, persistent antigen stimulation by the tumor results in the depletion of various populations of T-lymphocytes and increases the rate of their apoptosis [7,18] This is one of the mechanisms of tumor evasion from the immune response (immune evasion) [19]. Kim C.G. et al. showed that a decrease in the content of PD-1^+^CD8^+^ T-lymphocytes in the blood of patients with non-small cell lung cancer is associated with a better response to therapy with PD-1 inhibitors [20]. There is a need for therapy aimed at increasing the level and activity of CD8^+^ T-lymphocytes. Cells with properties of memory T-cells have been found in some subpopulations of T-lymphocytes [21]. This cell population is characterized by the expression of defined markers, including CCR7 [21]. Reprogramming of terminally differentiated lymphocytes that give them stem cell-like properties could be used to enhance the efficacy of immunotherapy against cancer and CSCs. We hypothesized that inhibiting immune checkpoints in vitro may be a promising approach to increase the antitumor activity of CD8^+^ T-lymphocytes in SCLC.

The aim of this study was to monitor CTCs and individual subpopulations of T-lymphocytes in the blood of patients with SCLC in order to identify new potential diagnostic markers. Additionally, evaluation of the effectiveness of cell therapy with reprogrammed CD8^+^ T blood lymphocytes on CSCs obtained from the blood of a patient with SCLC was carried out in vitro.

## 2. Results

### 2.1. A Significant Number of Different Subpopulations of CTCs and CSCs Are Present the Blood of Patients with SCLC

Tumor markers are expressed by cancer and by normal cells [22]. We compared the expression of tumor and stem markers in the blood cells of volunteers and patients with SCLC. The analysis of these data showed a significant increase in the number of tumor cells in the blood of patients with SCLC in comparison with healthy volunteers. These cells include CTCs with phenotypes CD87^+^, CD87^+^CD117^+^, CD117^+^Axl^+^EGF^+^CD44^+^, EGF^+^Axl^+^, CD117^+^EGF^+^, Axl^+^CD117^+^EGF^+^, CD276^+^, CD276^+^CD117^+^, CD87^+^CD276^+^, and EGF^+^Axl^−^ (Figure 1).

Moreover, the number of ALDH^+^ cells was increased in the blood of patients with SCLC. However, differences between the individual groups were not significant (Figure 2).

Expression of the SOX2 protein is associated with more aggressive tumors [23]. There is an association between SOX2 expression and the degree of SCLC, as well as with the survival of patients with SCLC [24]. An increase in SOX2 activity enhances the proliferation of tumor cells. In addition, overexpression of SOX2 is important for the function of the pulmonary CSCs [25]. In this regard, we evaluated the expression of SOX2 in blood mononuclear cells expressing tumor markers such as AXL, EGF, CD87, CD90, or CD276. Figure 3 shows that the number of cells with the phenotype Axl^+^SOX2^+^, EGF^+^SOX2^+^, and CD87^+^SOX2^+^ is significantly higher in patients with SCLC than in healthy volunteers (Figure 3).

### 2.2. Smoking Increases the Expression of Tumor Markers in Healhy Volunteers and the Number of CTCs in Patients with SCLC

Smoking is a risk factor for SCLC [26]. We evaluated the effect of smoking on the number of blood mononuclear cells expressing tumor markers. Our analysis revealed that the percentages of ALDH^+^ and CD276^+^ cells were higher in smoking volunteers than non-smokers (Figure 4a). The number of tumor cells with phenotype CD87^+^CD117^+^, CD117^+^Axl^+^EGF^+^CD44^+^, EGF^+^Axl^+^, CD117^+^EGF^+^, CD276^+^CD117^+^, and EGF^+^Axl^−^ had a tendency to increase.

Comparison of CTCs number in smokers and non-smokers with SCLC revealed group-specific differences. The number of ALDH^+^, EGF^+^Axl^−^, and CD87^+^CD276^+^ CTCs was increased in smokers with SCLC compared to non-smokers (Figure 4b). On the other hand, the number of CD276^+^ and CD276^+^CD117^+^ CTCs was lower in smokers with SCLC than in non-smokers with SCLC.

### 2.3. The Number of CD3^+^CD8^+^ T-Lymphocytes was Decreased in the Blood of Patients with SCLC

CD3^+^CD8^+^ T-lymphocytes are cytotoxic lymphocytes involved in the antitumor response. The percentage of CD3^+^CD8^+^ T-lymphocytes was decreased in the blood of patients with SCLC in comparison with healthy volunteers (Figure 5). 

The population of CD3^+^CD8^+^ T-lymphocytes is heterogeneous. Characteristics of the relevant cell markers are presented in the Appendix A. In the present study, we assessed the content of CD3^+^CD8^+^T-lymphocytes additionally expressing CD69, EGF, and Axl. According to our data, the number of activated CD3^+^CD8^+^CD69^+^ T-lymphocytes, proliferating CD3^+^CD8^+^CD69^+^Ki67^+^ T-lymphocytes, and CD3^+^CD8^+^CD69^+^Axl^+^ T-lymphocytes is decreased in patients with SCLC compared with healthy volunteers (Figure 5). We found that in patients with SCLC, there is an increase in the content of effector CD3^+^CD8^+^CD69^+^ and CD3^+^CD8^+^ T-lymphocytes expressing EGF relative to healthy volunteers.

### 2.4. Smoking Increases the Number of CD3^+^CD8^+^ T-Lymphocytes in Patients with SCLC

We compared the content of CD3^+^CD8^+^ T-lymphocytes in the blood of smokers and non-smokers with SCLC. The total population of CD3^+^CD8^+^ T-lymphocytes was significantly higher in smokers with SCLC than in non-smokers with SCLC (Figure 6). The same pattern was observed for subpopulations of T-lymphocytes with the immunophenotype CD3^+^CD8^+^CD69^+^, CD3^+^CD8^+^CD69^+^EGF^+^, and CD3^+^CD8^+^CD69^+^Axl^+^ (Figure 6).

### 2.5. Cytostatic-Resistant CTCs Are Present in Patients with SCLC after Chemotherapy

Chemotherapy (CT) reduced the populations of CTCs with the phenotype ALDH^+^, CD87^+^, CD87^+^CD117^+^, EGF^+^Axl^+^, and CD276^+^CD117^+^ in the blood of patients with SCLC in compared to patients before the treatment (Figure 7). However, cytostatic agents did not cause significant changes in the numbers of CD117^+^Axl^+^EGF^+^CD44^+^, CD117^+^EGF^+^ CTCs, Axl^+^CD117^+^EGF^+^, and CD276^+^ CTCs.

### 2.6. Chemotherapy had Differential Effects on CD3^+^CD8^+^ T-Lymphocytes in Patients with SCLC

CT caused an increase in the total population of CD3^+^CD8^+^ T-lymphocytes and subpopulations of T-lymphocytes with phenotypes CD3^+^CD8^+^CD69^+^ and CD3^+^CD8^+^CD69^+^Axl^+^ in the blood of patients with SCLC compared with those before treatment (Figure 8). However, cytostatic agents reduced subpopulations of T-lymphocytes with the phenotype CD3^+^CD8^+^CD69^+^EGF^+^ and CD3^+^CD8^+^EGF^+^ and did not influence CD3^+^CD8^+^Ki67^+^ T-lymphocytes relative to the values before CT.

### 2.7. Characteristics of CTCs and CD3^+^CD8^+^ T-Lymphocytes in Smoking Patient G with SCLC

A smoking patient G was selected from the general group of patients with SCLC (Appendix A). Initially, cytometry analysis of CTCs and CD3^+^CD8^+^ T-lymphocytes was performed in comparison with the general group of patients with SCLC. A non-smoking volunteer K was selected from the general group of volunteers (Appendix A). Next, the cytotoxic activity of rTcells of volunteer K was studied on primary culture of patient G derived tumor cells.

#### 2.7.1. A Larger Number of CTCs were Found in the Blood of Patient G Than in the General Group of Patients with SCLC

In the blood of patient G a larger number of CTCs with phenotypes ALDH^+^, Axl^+^CD117^+^EGF^+^, CD117^+^Axl^+^EGF^+^CD44^+^, EGF^+^Axl^+^, CD117^+^EGF^+^, Axl^+^SOX2^+^, and CD90^+^SOX2^+^ was found compared to the general group of patients with SCLC (Figure 9). The content of CTCs with phenotype CD87^+^, CD276^+^, CD87^+^CD117^+^, CD276^+^CD117^+^, EGF^+^SOX2^+^, CD87^+^SOX2^+^, and CD276^+^SOX2^+^ was as high as in the general group of patients with SCLC.

#### 2.7.2. A Larger Number of CD3^+^CD8^+^CD69^+^Axl^+^ and CD3^+^CD8^+^EGF^+^ T-Lymphocytes were Found in the Blood of Patient G compared to the General Group of Patients with SCLC

Quantitative parameters of T-lymphocytes of patient G did not differ from those in the general group: the number of cells with phenotypes CD3^+^CD8^+^, CD3^+^CD8^+^Ki67^+^, CD3^+^CD8^+^CD69^+^, and CD3^+^CD8^+^CD69^+^Ki67^+^ was reduced, while the number of CD3^+^CD8^+^CD69^+^EGF^+^ T cells was increased (Figure 10). CD3^+^CD8^+^CD69^+^Axl^+^ and CD3^+^CD8^+^EGF^+^ T-lymphocytes were an exception where the percentage was increased compared to the group of volunteers and the general group of SCLC patients.

#### 2.7.3. A Higher Number of Subpopulations of CTCs Resistant to Cytostatic were Detected in the Blood of Patient G

After chemotherapy there was a decrease in the number of CTCs with phenotypes ALDH^+^ (by 86.3%), CD117^+^Axl^+^EGF^+^CD44^+^ (by 48%), CD117^+^EGF^+^ (by 48%), and EGF^+^Axl^+^ (by 41%) in the blood of patient G relative to same indicators before treatment (Figure 11). Changes in the content of CTCs expressing CD87 and CD276 were not significant.

When studying the expression of the SOX2, it was found that CT caused a significant reduction in the subpopulation of CD90^+^SOX2^+^ CTCs (by 63%) in the blood of patient G in comparison with the values before treatment (Figure 12). At the same time, cytostatics did change the percentage of EGF^+^SOX2^+^ and CD276^+^SOX2^+^ CTCs, while the number of CD87^+^SOX2^+^ CTCs increased by 53%.

#### 2.7.4. Differences in the Effect of Cytostatics on CD3^+^CD8^+^ T-Lymphocytes of Patient G were Revealed Relative to the General Group of Patients with SCLC

CT increased the total population of CD3^+^CD8^+^ T-lymphocytes (by 20%) in the blood of patient G relative to those before treatment. At the same time, there was an increase in the number of T-lymphocytes with phenotypes CD3^+^CD8^+^CD69^+^ (by 67%), CD3^+^CD8^+^CD69^+^EGF^+^ (by 25.5%), and CD3^+^CD8^+^CD69^+^Axl^+^ (by 49%) (Figure 13). However, the number of CD3^+^CD8^+^EGF^+^ T-lymphocytes decreased significantly (by 85.5%). Changes in CD3^+^CD8^+^Ki67^+^ T-lymphocytes and CD3^+^CD8^+^CD69^+^Ki67^+^ T-lymphocytes were not significant.

#### 2.7.5. Tumor Cells Isolated from the Blood of Patient G Form Spheroids In Vitro, Which Included Cells Expressing CD87, CD117, CD274, EGF, and Axl

When studying CTCs isolated from the blood of patient G in vitro, we found that cells in Matrigel form spheroids and conglomerates of several spheroids with an average size of up to 100 µm (Table 1). 

A spheroid was defined as a three-dimensional cellular structure (Figure 14). Spheroids were divided into three classes by cellularity. In total, 64 small spheroids containing 10–14 cells were found in the culture, 31 medium spheroids containing 15–19 cells were found, and 6 large spheroids containing 20–34 cells were found. The cells within spheroid were larger than cells on the outside. In spheroid cells, the Hoechst 34580 nuclear dye was evenly distributed. Most often, the spheroids stained with Hoechst 34580 less intensely than the cells on the outside. Dead cells were not found in the structure of spheroids.

When evaluating cells in culture, we used different combinations of antibodies: Hoechst/CFSE/EGF, CD87/CD117/EGF, Axl/CD117/EGF, and CD274/CD117/EGF. Immunofluorescence revealed 100% staining of spheroid cells with antibodies to CD87, Axl, CD117, and EGF. In addition, the tumor marker CD274 (PD-L1) was found on spheroid cells.

Additionally, single cells located in Matrigel were evaluated. When cells were stained in the Hoechst/CFSE/EGF sequence, it was revealed that the nuclei of all cells (100%) were stained with Hoechst 34580, while 55.14% of the cells were labelled with EGF and 47.20% of the cells were active for CFSE. All CFSE-stained cells were also stained with Hoechst 34580 and EGF (Figure 15). When cells were stained for the CD87/CD117/EGF, it was found that 76.28% of EGF^+^ cells expressed CD117, and 53.09% of EGF^+^ cells expressed CD87. All CD87^+^ cells were positive for CD117 and EGF (Figure 15). When cells were stained for Axl/CD117/EGF, 68.29% of Axl-positive cells were positive for CD117 and EGF (Figure 15). When cells were stained for CD274/CD117/EGF, 53.56% of CD274^+^ cells were positive for CD117 and EGF (Figure 15).

### 2.8. Reprogrammed CD3^+^CD8^+^ T-Lymphocytes of Volunteer K Have Cytotoxic Effects on CSC of Patient G In Vitro

The efficiency of reprogramming was studied by evaluating the expression of CCR7 by T-lymphocytes (Figure 16). Evaluation of the cytotoxic activity of rTcells was carried out in comparison with naive CD3^+^CD8^+^ T-lymphocytes (nTcells) in the primary culture of CSCs isolated from the blood of patient G (Figure 17, Table 2). rTcells had a higher survival rate in culture of CSCs than nTcells. This pattern was observed in the ratio of T-lymphocytes and CSCs of 0.25:1, 1:1, 5:1, and 10:1.

At all concentrations, rTcells and nTcells had a cytotoxic effect on CSCs. With an increase in the concentration of rTcells and nTcells, the number of dead CSCs increased, and the maximum cytotoxic effect was observed at a ratio of lymphocytes and CSCs of 10:1 (Table 2). At the same time, the cytostatic effect of rTcells was superior to that of nTcells in cell ratios of 5:1 and 10:1.

Additionally, CSCs were co-cultivated with rTcells. In the ratio of 0.25:1 and 1:1, rTcells formed small clusters (up to 5–6) around spheroids and single CSCs. At the same time, stained rTcells were not detected around singly located 7-AAD cells and spheroids.

At a ratio of rTcells/CSCs of 2.5:1, lymphocytes formed clusters (up to 6–10) around spheroids and solitary CSCs. Stained rTcells were not detected around singly located 7-AAD cells and spheroids.

With an increase in the ratio of rTcells/CSCs to 5:1, lymphocytes accumulate in large numbers around spheroids (50–60% of the spheroid surface is covered with lymphocytes) and single CSCs. Stained rTcells were not detected around singly located 7-AAD cells and spheroids.

At a ratio of rTcells/CSCs of 10.0:1.0, lymphocytes formed clusters around spheroids (80–90% of the spheroid surface is covered with lymphocytes) and singly located CSCs. The stronger the color intensity of 7-AAD rTcells, the lower their number was found on the surface of spheroids. Stained rTcells were not detected around singly located 7-AAD cells and spheroids.

By interaction with rTcells, spheroids were divided into several groups:−Spheroids without apoptotic changes without the inclusion of rTcells;−Spheroids with apoptotic changes with the inclusion of rTcells;−Spheroids with apoptotic changes without the inclusion of rTcells.

As shown in Table 3, the greatest cell apoptosis was observed in spheroids with rTcells inclusion in the ratios rTcells/CSCs 1:1, 2.5:1, 5:1, and 10:1 and in spheroids without rTcells inclusion in the ratios rTcells/CSCs 10:1 (Table 3).

## 3. Discussion

SCLC is the most aggressive form of cancer. It is characterized by a long, asymptomatic course and a poor prognosis. Despite the successes, early diagnosis and dynamic monitoring of therapeutic efficacy of SCLC, as well as the strengthening of already known approaches to the treatment, efficacy of the therapy remains modest. Due to resistance to chemotherapy, the 5-year survival rate of patients with SCLC remains low [4,27]. Recently, some progress has been achieved in understanding causes of tumor resistance to chemotherapy and establishing the mechanisms of tumor escape from the immune response. According to available data, SCLC resistance to chemotherapy, metastasis, and recurrence of the disease is associated with CSCs, a population for which the identification of targets and biomarkers is still under development. CSCs are part of the CTCs population and characterized by their ability to form spheroids in vitro and larger multicellular structures called tumorspheres [16,28].

In the present study, we evaluated the levels of CTCs in the blood of patients with SCLC. In the general population of patients with SCLC, there was a significant increase in CD87^+^, CD87^+^CD117^+^, and CD276^+^CD117^+^ (Figure 1). The proportion of other investigated cell populations tended to increase (Figure 1 and Figure 2). We propose that the analysis of CTCs may represent a useful tool to monitor the evolution of malignancies, especially when tumor biopsy is uncomfortable for the patient or totally impracticable, as it often is in lung cancer.

It is known that SOX2 is associated with the severity of SCLC [26]. We evaluated the expression of SOX2 on blood cells expressing Axl, EGF, CD87, CD90, or CD276. It turned out; the number of cells expressing SOX2 is significantly higher in patients with SCLC than in the group of healthy volunteers (Figure 3). Thus, a number of tumor cells and CTCs (CD87^+^, CD87^+^CD117^+^, and CD276^+^CD117^+^ and Axl^+^SOX2^+^, EGF^+^SOX2^+^, and CD87^+^SOX2^+^) may be interesting as diagnostic markers of SCLC or indicators of dynamic disease control.

In the vast majority of cases, SCLC is associated with smoking [26]. It is estimated that 90% of lung cancers to be attributable to smoking, and smokers have as much as a 30-fold increased risk of developing cancer [29]. Carcinogens contained in cigarette smoke are able to induce the formation of CSCs and increase the expression of CSC markers [29]. In the present study, the number of CTCs EGF^+^Axl^+^, CD87^+^, and CD87^+^CD117^+^ had increasing tendency in the blood of smoking volunteers. Interestingly, more CTCs with ALDH^+^ phenotype were detected in smoking patients with SCLC than in non-smoking patients with SCLC (Figure 4). At the same time, the clinical parameters of smokers are significantly worse than those of non-smoking patients. 

This observation requires attention, since, in our opinion, an increase in the expression of Axl, CD87, and CD117 in smokers is the basis for a detailed examination in order to early diagnose malignant neoplasms of lungs, in particular SCLC. However, we understand that obtained results are not enough for the final recommendations, it is necessary to confirm our findings in a larger sample of patients.

CD8^+^ T-lymphocytes are one of the most important effector cells in antitumor response. However, lymphopenia is frequent in patients with cancer, and T-lymphocytes become dysfunctional under the influence of tumor and tumor environment [7,18]. Impairment of the immunological function of these cells is one of mechanisms for the development and progression of lung cancer [30]. We evaluated the content of different populations of cytotoxic T-lymphocytes in the blood of patients with SCLC. The total population of cytotoxic T-lymphocytes and individual subpopulations of T-lymphocytes were decreased (Figure 5). On the contrary, the content of CD3^+^CD8^+^Ki67^+^ and CD3^+^CD8^+^EGF^+^ T-lymphocytes was increased. It is believed that increase of EGF receptors expression is associated with an improvement in the antitumor activity of CD8 positive T-lymphocytes. Some reports suggest that in the presence of EGFR ligand secreted by tumor cells, CD3^+^CD8^+^CD69^+^Ki67^+^ T-lymphocytes produce more IFN-γ and TNF-α and show a stronger antitumor response [27]. Due to this mechanism, they can delay tumor growth in vivo [27]. Together with an increase in the number of cells carrying the Ki67 cell proliferation marker may be a consequence of a compensatory response to a decrease in the total population of CD3^+^CD8^+^ T-lymphocytes.

Different independent research groups have shown high heterogeneity of tumor cells in various types of cancers [31,32,33,34]. We confirmed this for SCLC. The plasticity and heterogeneity of tumor cells contributes to the selection of resistant populations of CSCs from the general population of tumor cells in response to radiation and chemotherapy [35,36,37]. In this regard, the identification of CT-resistant populations of CTCs in patients with SCLC may be helpful in predicting the disease and treatment correction. 

Our results showing a differential sensitivity of CTCs to the effects of chemotherapeutic agents in patients with SCLC are generally consistent with previously published data showing a controversial response of CSCs in lung cancer and other malignant neoplasms (Figure 7) [15,38,39,40]. On the other hand, the revealed subpopulations of CTCs resistant to cytostatics can be attributed to pathogenetic factors of tumor progression and its metastasis.

Chemotherapy does not only affect tumor cells and CSCs. As shown in Figure 8, the population of CD3^+^CD8^+^ T-lymphocytes of the blood of patients with SCLC increased after chemotherapy. This was mainly due to an increase in the number of activated T-lymphocytes (CD3^+^CD8^+^CD69^+^). Positive changes in cytotoxic CD3^+^CD8^+^ T-lymphocytes, at least in the proliferative activity of immune system cells can be explained by a decrease in the negative effect of the tumor. However, clinical observations did not reveal positive changes in the condition of the patients included in this study. We associate this to the persistence of the negative effects of the tumor on the function of immune cells. Cell therapy with CD3^+^CD8^+^ T-lymphocytes targeted to CSCs and more resistant to the negative effects of the tumor can be a possible solution of the issue. Previously, we demonstrated an increased cytotoxicity and feasibility of using reprogrammed murine CD3^+^CD8^+^ T-lymphocytes to eliminate CSCs in Lewis lung carcinoma in mice [41]. We hypothesized that reprogramming of CD3^+^CD8^+^ T-lymphocytes may increase the cytotoxic activity of these cells against CSCs of patients with SCLC.

We evaluated in vitro cytotoxic effects of reprogrammed T-lymphocytes in a culture of CSCs isolated from the blood of a patient with SCLC (patient G). CD3^+^CD8^+^ T-lymphocytes were obtained from the blood of volunteer K subjected to reprogramming. Reprogramming was achieved by inhibition of the MAPK/ERK pathway via MEK1/2i and blockade of the PD-1/PD-L1 signaling pathway by human monoclonal antibody nivolumab.

Using a cytometry analysis, a greater number of CTCs were detected in the blood of patient G than in the general group of patients with SCLC (Figure 9). After a course of chemotherapy, ALDH^+^, CD117^+^Axl^+^, CD117^+^Axl^+^EGF^+^CD44^+^, CD117^+^EGF^+^, and EGF^+^Axl^+^ were slightly decreased, which corresponded to the general trend found in the general group of examined patients with SCLC. Changes in the content of T-lymphocytes were comparable to those found in the general group (Figure 10).

In vitro, we assessed the sphere formation capability of the CTCs obtained from patient G and the structure of the spheroids. It was found that CTCs formed spheroids and conglomerates of spheroids up to 100 µm in size. After 14 days of cultivation, 101 spheroids were obtained, and the number of cells in spheroids ranged from 11 to 32. Spheroids with cellularity from 10 to 14 cells predominated (63.37% of the total number of spheroids) (Table 1). Staining of single cells showed that they expressed the same markers that were previously identified in the blood of a patient with SCLC (CD274, CD117, Axl, and EGF) (Figure 15). Sphere formation is considered to be a characteristic feature of CSCs [16,28]. Thus, our data confirm the presence of CSCs in the general population of CTCs obtained from patient G. At the final stage of our study, we assessed antitumor activity of rTcells of volunteer K blood rTcells in the culture of CSCs of the patient G blood. During reprogramming of CD3^+^CD8^+^ T-lymphocytes of volunteer K, tumor cells of patient G were used to prepare an antigen-presenting mixture. Thus, a targeted cytotoxic effect of rTcells on CSCs was achieved in vitro. The rTcells were characterized by a high level of CCR7 expression (Figure 16). This indicates that rTcells exhibit properties of memory T-cells. In addition, rTcells were significantly more resistant to the cytotoxic effect of tumor cells than naive CD3^+^CD8^+^ T-lymphocytes. Naive CD3^+^CD8^+^ T-lymphocytes and rTcells showed cytotoxic properties against tumor cells. However, antitumor activity of rTcells was higher than those of naive CD3^+^CD8^+^ T-lymphocytes.

We recognize that the present study has some limitations. We conducted a short-term, single-center study with a relatively small sample size, which reduces the likelihood of generalization. In order to evaluate the applicability of the research results to a larger population, the data presented above need further validation using multicenter cohorts with large numbers of patients.

## 4. Material and Methods

### 4.1. Patients

The study included 7 patients with IIC–IV SCLC, aged from 45 to 75 years (average age of 56.6 ± 1.3 years) who received treatment at the Cancer Research Institute of Tomsk NRMC (Tomsk, Russia) from 2021 to 2022. Histological diagnosis was confirmed for all samples (Appendix A). All patients received standard chemotherapy (atezolizumab, etoposide, combination of platinum drugs). Four out of seven patients had COPD, and four patients reported being a smoker. Blood samples were obtained from patients before and after the first course of chemotherapy. Blood samples from 8 healthy volunteers of similar age were used as control.

This study was a pilot investigation. Informed consent was obtained from all individual participants. All procedures involving human participants were conducted in accordance with the ethical standards of the institutional and/or national research committee and with the 1964 Helsinki Declaration and its later amendments or comparable ethical standards.

### 4.2. Design of Investigation

The study design is shown in Figure 18. At the first stage, the content of CTCs and various populations of CD8^+^ T-lymphocytes obtained from healthy volunteers and patients with SCLC were assessed in blood samples. Mononuclear cells were isolated from the blood of a patient with SCLC. Cells were cultured, and an adherent fraction of mononuclear cells was used for further experiments. Adherent fraction of mononuclear cells from a patient with SCLC was divided into 3 parts: part 1 was used to obtain an antigen-presenting mixture, part 2 was used to analyze the cytotoxicity of CD8^+^ T-lymphocytes, and part 3 was used to differentiate staining of cultured spheroids. CD8^+^ T-lymphocytes were isolated from the blood of a healthy volunteer using magnetic separation and were subsequently reprogrammed. In the following, we assessed cytotoxic activity of reprogrammed CD8^+^ T-lymphocytes in the CSCs culture in vitro. Additionally, we studied apoptosis of reprogrammed CD8^+^ T-lymphocytes.

### 4.3. Isolation of Blood Mononuclear Cells

Lympholyte-H (CEDARLANE, Netherlands, Cedarlane Laboratories, Cat#CL5015) protocol was used for the elimination of erythrocytes and dead cells from human blood and receiving mononuclear cells.

### 4.4. Cryopreservation of Mononuclear Cells Obtained from Blood of Patients with SCLC

For cryopreservation of mononuclear cells obtained from blood of patients with SCLC and healthy volunteers, CryoStor^®^ CS5 cryopreservation medium (serum-free, containing 5% dimethyl sulfoxide (DMSO) StemCell Technologies, Kent, WA, USA) was used. Cold (2–8 °C) CryoStor^®^ CS5 was added to the cell suspension at a rate of 10 million cells/1 mL. The suspension was thoroughly mixed and placed in a cryovial. Cells were incubated at 2–8 °C for 10 min. The slurry was then cooled using a controlled slow rate cooling protocol (1 °C/minute) and stored at liquid nitrogen temperature (−135 °C). Cells were thawed in a water bath at 37 °C and culture medium RPMI 1640 (Sigma-Aldrich, St. Louis, MO, USA) warmed to 37 °C was added with 10% fetal bovine serum (FBS, Sigma-Aldrich, St. Louis, MO, USA), 10 mM HEPES (Sigma-Aldrich, St. Louis, MO, USA), and 55 µM β-mercaptoethanol (Thermo Scientific™ 35602BID, Thermo Scientific, Waltham, MA, USA) in the sample: culture medium = 1:10. The suspension was washed twice at 300 g for 10 min at room temperature (15–25 °C).

### 4.5. Magnetic Separation of Human CD8^+^ T-lymphocytes

After isolation of mononuclear cells from blood, a magnetic separation was performed to enrich the cell fraction with CD8^+^ T-lymphocytes. Enrichment was performed following a standard protocol using a commercial kit (EasySep^TM^ Human CD8^+^ T Cell Isolation Kit), as recommended by the manufacturer (StemCell Technologies, Vancouver, BC, Canada).

### 4.6. Flow Cytometry

Mononuclear cells from blood were isolated as described above, and the expression of surface markers on mononuclear cells was analyzed using flow cytometry. Fc-receptors were blocked by pre-incubation of the cells with unconjugated anti-CD16/CD32 antibodies for 10 min (eBioscience, San Diego, CA, USA, Cat# 464219, 1/100 dilution) in 50 μL of 0.1% saponin (Sigma-Aldrich, St. Louis, MO, USA, Cat# S4521) and 1% bovine serum albumin (BSA, Sigma-Aldrich, St. Louis, MO, USA, Cat# A3059-100G) in phosphate-buffered saline (PBS, Sigma-Aldrich, St. Louis, MO, USA) per tube. After the pre-incubation, cells suspensions were stained with fluorophore-conjugated monoclonal antibodies. CSCs and T-lymphocyte subpopulations were defined according to the markers specified in Appendix A and using different combinations of the following monoclonal antibodies anti-: CD3 PerCP, CD8 FITC, CD44 APC-H7, CD69 PE, CD87 BV421, CD117 BB515, CD276 BV750, Axl BV480, and EGF Receptor Alexa Fluor^®^ 647 (all Becton Dickinson, San Jose, CA, USA). We used BD FastImmune reagent as a combination of CD8 FITC, CD69 PE, and CD3 PerCP (1/5 dilution, Cat# 340367). The monoclonal antibodies CD44 APC-H7 (QC Testing: Human, Clone: G44-26, Cat#560532, 1/10 dilution), CD87 BV421 (QC Testing: Human, Clone: VIM5, Cat#743095, 1/10 dilution), CD90 APC (QC Testing: Human, Clone: 5E10, Cat#743095, 1/10 dilution), CD117 BB515 (QC Testing: Human, Clone: 104D2, Cat#565172, 1/10 dilution), CD276 BV750 (QC Testing: Human, Clone: 7-517, Cat# 746976, 1/10 dilution), Axl BV480 (QC Testing: Human, Clone: 108724, Cat#747863, 1/10 dilution), and EGF Receptor Alexa Fluor^®^ 647 (QC Testing: Human, Clone: EGFR.1, Cat#563577, 1/10 dilution) were used. All antibodies were titrated to determine their optimal staining concentration and appropriate isotype controls were used. Further, the cell suspension was stained with Ki-67Alexa Fluor^®^ 647 intracellular antibodies (1/10 dilution, Clone B56, Cat#558615, BD Biosciences, San Jose, CA, USA) or SOX2 Alexa Fluor 488 (1/5 dilution, Clone 030-678, Cat#561593, BD Biosciences, San Jose, CA, USA). Labelled cells were washed thoroughly with 500 μL of FACSFlow (Becton Dickenson, Franklin Lakes, NJ, USA, Cat# 342003). The desired CSCs and T-lymphocyte subpopulations were gated using doublet discrimination and forward and side scatter, and their relative percentages were determined. The flow cytometry data were analyzed using FACSDiva software (BD Biosciences).

### 4.7. Measurement of ALDH Activity

An aldehyde dehydrogenase-based cell detection kit (StemCell Technologies, Vancouver, BC, Canada, Cat #01700) was used to determine ALDH1 enzymatic activity in the blood. Cells were suspended in aldefluor assay buffer and incubated with the ALDH enzyme substrate, BODIPY-amino acetaldehyde (BAAA), for 40 min at 37 °C. As a control, cells were also treated with diethylaminobenzaldehyde (DEAB), an inhibitor of ALDH enzyme activity. Fluorescence was determined using a BD FACS Canto II flow cytometry and analyzed using FACSDiva software (BD Biosciences).

### 4.8. Cultivation of CSCs Obtained from Blood of Patients with SCLC

Cells of the adherent fraction of blood mononuclear cells of a patient with SCLC were counted, mixed with cell culture medium (RPMI 1640 (Sigma-Aldrich, St. Louis, MO, USA) supplemented with 10% FBS (Sigma-Aldrich, St. Louis, MO, USA), 2 mM L-glutamine (Sigma-Aldrich, St. Louis, MO, USA)) containing 0.5 mg/mL Matrigel (Corning^®^ Matrigel^®^ Basement Membrane Matrix High Concentration, Sigma-Aldrich, St. Louis, MO, USA) [42]. The final concentration of cells in the medium is 2 × 10^5^/mL. Then 100 µL of the mixture was placed in 96-well plates pre-coated with 50 µL of 1% agarose low EEO (Acros Organics, Geel, Belgium). After 2 h, the cell layer was covered with 100 μL of cell culture medium without Matrigel. Cells were cultured for 14 days. The culture medium was changed every other day. For that, the medium was carefully removed without touching the cell layer, and a new cell culture medium was added. Cell clusters were scored and counted automatically as described below using a Cytation 5 Cell Imaging multi-mode reader (BioTek Instruments, Inc., Winooski, VT, USA) using Gen5 software (BioTek, Instruments, Friedrichshall, Germany).

For automatic analysis of survival and proliferation, each well was visualized at least 7 different focal planes. Focus summation was applied to Z-level images to obtain one clear image of all 3D colonies. Further processing included background subtraction, median filtering, and thresholding. After that, an automatic count of colonies was carried out, displaying the number of colonies, the average area of colonies, and the total area of colonies.

### 4.9. Reprogramming of Human CD8^+^ T-Lymphocytes

After enrichment of the cell suspension with CD8^+^ T-lymphocytes using magnetic separation, the cells were incubated in the medium recommended for CD8^+^ T-lymphocytes (RPMI 1640 (Sigma-Aldrich, St. Louis, MO, USA) with the addition of 10% FBS (Sigma-Aldrich, St. Louis, MO, USA), 2 mM L-glutamine (Sigma-Aldrich, St. Louis, MO, USA), 10 mM HEPES (Sigma-Aldrich, St. Louis, MO, USA), and 55 μM β-mercaptoethanol (Thermo Scientific™ 35602BID, Thermo Scientific, Waltham, MA, USA), 37 °C, 5% CO_2_) for 2–3 h. The concentration of T-lymphocytes was 1 × 10^8^/mL. The volume of the medium in the vial was at least 5 mL.

Cells isolated from the adherent fraction of mononuclear cells of a patient with SCLC were cultured for 10 days as described above. The cells were harvested from plastic and washed twice by centrifugation, and then an antigen-presenting mixture was prepared. An antigen-presenting mix was prepared from CSCs lysate by using a freeze-thaw cycle in 0.85% NaCl solution. The cycle was repeated five times in rapid succession from −70 °C to 37 °C, and then re-frozen and stored at −70 °C before the use. After the final thawing, the lysate was stained by trypan blue (Sigma-Aldrich, St. Louis, MO, USA) [41]. The preparation of adjuvant (Freund’s adjuvant) for the antigen-presenting mix was carried out according to the manufacturer’s standard protocol (Sigma-Aldrich, St. Louis, MO, USA). Freund’s adjuvant solution was mixed with the tumor cell lysate (3 × 10^4^/mL) at a 1:1 ratio to form a thick emulsion.

Human CD8^+^ T-lymphocytes were reprogrammed as described earlier [43]. Immunophenotype and cytotoxicity of reprogrammed CD8^+^ T-lymphocytes were analyzed using by Cytation 5 (BioTek Instruments, Inc., Winooski, VT, USA).

### 4.10. Detection of the CCR7 Expression on Reprogrammed Human CD8^+^ T-Lymphocytes In Vitro

Images of CD8^+^ T-lymphocytes and CSCs were obtained using the cell imaging Cytation 5 (BioTek Instruments, Inc., Winooski, VT, USA) equipped with the following cubes: DAPI (blue), GFP (green), and RFP (yellow). 

To assess CCR7 expression, reprogrammed human T-lymphocytes were stained with anti-CCR7 antibodies and polyclonal secondary antibody donkey anti-Rabbit IgG H&L Alexa Fluor^®^ 555 (all Abcam, Cambridge, MA, USA). Nuclei were additionally stained with Hoechst 34580 (blue); CD8 FITC (green) was used to CD8^+^ T-lymphocyte detection. The percentage of CD8^+^CCR7^+^ cells were determined as the ratio of cells counted in green and yellow channel to total cells counted using blue (DAPI) channel. 

All images were obtained using a Cytation 5 device (4× or 20× magnification) followed by cell analysis using Gen5™ data analysis software (BioTek, Instruments, Friedrichshall, Germany). Prior to the analysis, images were pre-processed to align the background.

### 4.11. Detection of the Cytotoxicity and Apoptosis of Reprogrammed Human CD8^+^ T-Lymphocytes In Vitro

Cytotoxicity and apoptosis of reprogrammed and naive CD8^+^ T-lymphocytes were studied in cell culture of CSCs isolated from adhesive fraction of mononuclear cells from patient with SCLC. After co-incubation, reprogrammed and naive human CD8^+^ T-lymphocytes and CSCs were stained with Hoechst 34580 (for nuclear staining, Becton Dickinson, San Jose, CA, USA) and 7-AAD (for apoptotic cells detection, Becton Dickinson, San Jose, CA, USA). Preliminary reprogrammed and naive human CD8^+^ T-lymphocytes were stained with CFSE during 24 h. CFSE staining of CD8^+^ T-lymphocytes was performed according to the manufacturer’s instruction (Becton Dickinson, San Jose, CA, USA). Cytotoxicity of CD8^+^ T-lymphocytes in CSCs culture was assessed by analysing the ratio of cells counted in the blue and yellow channels to the total number of CSCs (percentage of dead CSC Hoechst^+^7-AAD^+^). Determination of the percentage of dead CD8^+^ T-lymphocytes Hoechst^+^CFSE^+^7-AAD^+^ was made by the ratio of cells counted in blue and green channel to total cells.

All images were obtained with Cytation 5 (4× or 20× magnification) followed by cell analysis using Gen5™ data-analysis software (BioTek, Instruments, Friedrichshall, Germany). Prior to the analysis, images were pre-processed to align the background.

### 4.12. Detection of the CSCs

For confirmation of the presence of CSC markers on single cells and tumor spheroids formed during the cultivation of the adhesive fraction of mononuclear cells obtained from a patient with SCLC, staining with fluorophore-conjugated monoclonal antibodies was performed (CD87 BV421, CD117 BB515, CD274 BV421, Axl BV480, EGF Receptor Alexa Fluor^®^ 647, and CFSE, all Becton Dickinson, San Jose, CA, USA)). Spheroids and single cells were stained with Hoechst 34580 (for nuclear staining, Becton Dickinson, San Jose, CA, USA), CD87, CD117, CD274, Axl, EGF, and/or CFSE during 30 min. Dyes were used in various combinations (Hoechst/CFSE/EGF) (CD87/CD117/EGF), (Axl/CD117/EGF), (CD274/CD117/EGF). CSCs culture was assessed by analyzing the ratio of cells counted in the blue and/or green/yellow channels. Tumor spheroids were defined as a three-dimensional cellular structure based on images obtained using Cytation 5 (4× or 20× magnification) followed by cell analysis using Gen5™ data analysis software (BioTek, Instruments, Friedrichshall, Germany). Prior to the analysis, images were pre-processed to align the background.

Before staining analysis of single cells, tumor spheroids were excluded by size using the Gen5™ data-analysis software interface (BioTek, Instruments, Friedrichshall, Germany).

### 4.13. Statistical Analysis

All statistical analyses were carried out by using SPSS statistical software (version 15.0, SPSS Inc., Chicago, IL, USA). Statistical analysis was performed using the Mann–Whitney U test and Wilcoxon signed-rank test. Additionally, *p* values < 0.05 were considered to indicate statistically significant differences. All quantitative data presented are the mean value and standard error.

## 5. Conclusions

Based on the results of in vivo and in vitro study, we propose an algorithm for the study of tumor cells and CSCs and CD3^+^CD8^+^ T-lymphocytes in order to improve the diagnosis of SCLC and increase the efficiency of chemotherapy assessment and its enhancement using reprogrammed T-lymphocytes (Figure 19).

In the present study, we obtained data that demonstrate an approach to improving the diagnosis, assessment of antitumor immune system, and monitoring the effectiveness of chemotherapy in SCLC based on the dynamic control of the content of cytotoxic CD3^+^CD8^+^ T-lymphocytes, CSCs, and CTCs. We show the possibility of increasing the effectiveness of the existing therapy with rTcells, which target to CSCs.

Thus, the results of in vitro study indicate the effectiveness of reprogramming to achieve the resistance of CD3^+^CD8^+^ T-lymphocytes of volunteer’s blood to the cytotoxic effect of tumor cells of a patient with SCLC. This achieves the targeted cytotoxic effect of rTcells on CSCs with a certain phenotype: cells are positive for CD274, CD117, Axl, and EGF.

We understand that the proposed algorithm has limitations, exceptions, and is dependent on the stage of disease. However, our results might provide an opportunity to change diagnostic measures in order to increase the effectiveness of therapy. We are currently optimizing the algorithm and recruiting additional patients. Further studies will provide new insights into the tumorigenic process and further validate the potential of CSCs in SCLC as therapeutic targets and biomarkers.

## Figures and Tables

**Figure 1 ijms-23-10853-f001:**
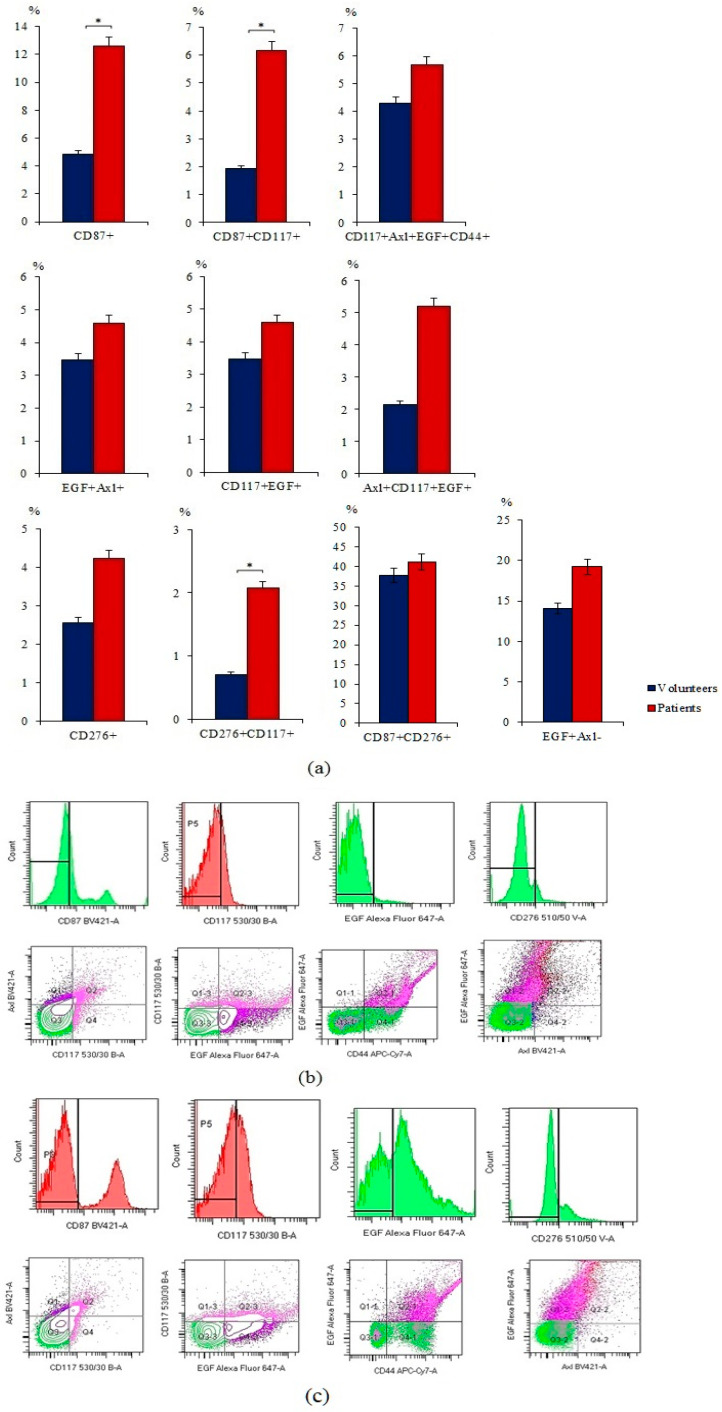
Characterization of circulating tumor cells (CTCs) isolated from the blood of patients with small cell lung cancer (SCLS) and healthy volunteers. Cells were analyzed by flow cytometry using antibodies against CD87, CD117, CD276, CD44, Axl, and EGF. (**a**) Levels of CD87^+^, CD87^+^CD117^+^, CD117^+^Axl^+^EGF^+^CD44^+^, EGF^+^Axl^+^, CD117^+^EGF^+^, Axl^+^CD117^+^EGF^+^, CD276^+^, CD276^+^CD117^+^, CD87^+^CD276^+^, and EGF^+^Axl^−^ CTCs in the blood of patients with SCLC and healthy volunteers (% of all labelled mononuclear cells); (**b**) phenotype establishment and qualitative analysis of CD44 (APC-H7), CD117 (BB515), CD87 (BV421), CD276 (BV510), EGF(AF647), and Axl (BV421) expression in blood of healthy volunteers; **(c)** phenotype establishment and qualitative analysis of CD44 (APC-H7), CD117 (BB515), CD87 (BV421), CD276 (BV510), EGF(AF647), and Axl (BV421) expression in blood of patients with SCLS. * Differences are significant in comparison with volunteers (*p* < 0.05).

**Figure 2 ijms-23-10853-f002:**
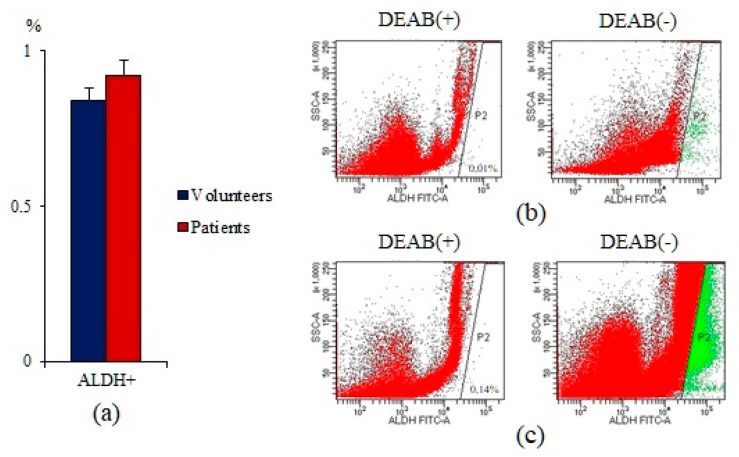
Cytometric analysis of circulating ALDH1^+^ cells from the blood of healthy volunteers and SCLC patients. (**a**) Level of circulating ALDH1^+^ cells in the blood of healthy volunteers and SCLC patients; (**b**) Aldefluor FACS analysis of circulating ALDH1^+^ cells in the blood of healthy volunteers; (**c**) Aldefluor FACS analysis of circulating ALDH1^+^ cells in the blood of the SCLC patients.

**Figure 3 ijms-23-10853-f003:**
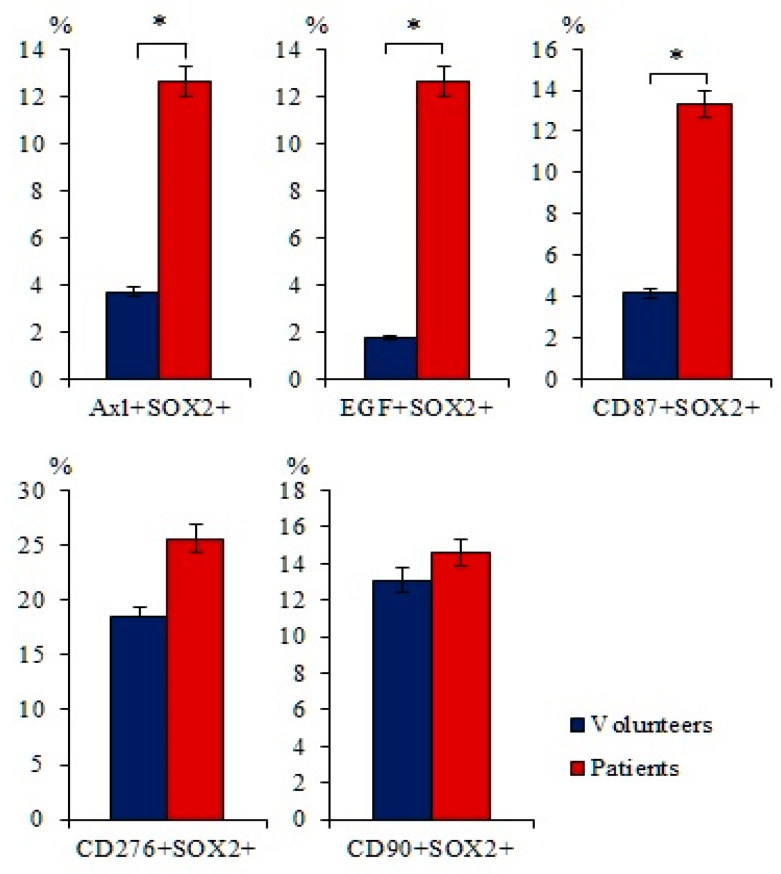
The content of Axl^+^SOX2^+^, EGF^+^SOX2^+^, CD87^+^SOX2^+^, CD276^+^SOX2^+^, and CD90^+^SOX2^+^ circulating tumor cells (CTCs) isolated from the blood of patients with SCLS and healthy volunteers (% of all labelled mononuclear cells). Cells were analyzed by flow cytometry using antibodies for Axl, EGF, CD87, CD276, CD90, and SOX2. * Differences are significant in comparison with volunteers (*p* < 0.05).

**Figure 4 ijms-23-10853-f004:**
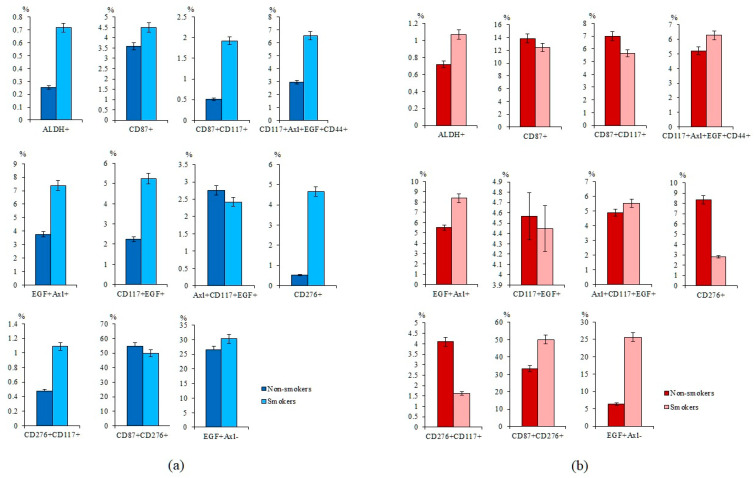
Characterization of circulating tumor cells (CTCs) isolated from the blood of non-smoking and smoking SCLC patients and healthy volunteers. Cells were analyzed by flow cytometry using antibodies against CD87, CD117, CD276, CD44, Axl, EGF, and ALDH1. The level of CD87^+^, CD87^+^CD117^+^, CD117^+^Axl^+^EGF^+^CD44^+^, EGF^+^Axl^+^, CD117^+^EGF^+^, Axl^+^CD117^+^EGF^+^, CD276^+^, CD276^+^CD117^+^, CD87^+^CD276^+^, and EGF^+^Axl^−^ CTCs in the blood of (**a**) non-smoking and smoking healthy volunteers and (**b**) non-smoking and smoking patients with SCLC (% of all labelled mononuclear cells).

**Figure 5 ijms-23-10853-f005:**
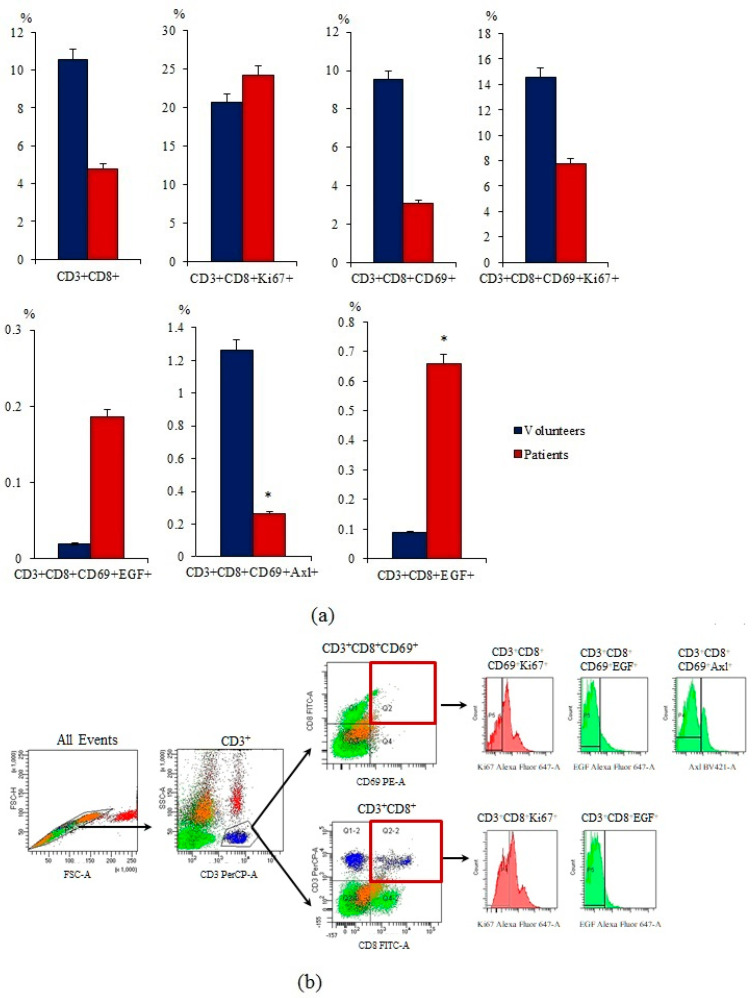
Characterization of CD3^+^CD8^+^ T-lymphocytes in the blood of healthy volunteers and SCLC patients. (**a**) The content of CD3^+^CD8^+^, CD3^+^CD8^+^Ki67^+^, CD3^+^CD8^+^CD69^+^, CD3^+^CD8^+^CD69^+^Ki67^+^, CD3^+^CD8^+^CD69^+^EGF^+^, CD3^+^CD8^+^CD69^+^Axl^+^, and CD3^+^CD8^+^EGF^+^ in the blood of healthy volunteers and SCLC patients (% of all labelled mononuclear cells); (**b**) cells were analyzed by flow cytometry using antibodies against CD3, CD8, CD69, EGF, Axl, and Ki67. Gating strategy for cytometry analysis of subpopulations from CD3^+^CD8^+^ T-lymphocytes. * Differences are significant in comparison with volunteers (*p* < 0.05).

**Figure 6 ijms-23-10853-f006:**
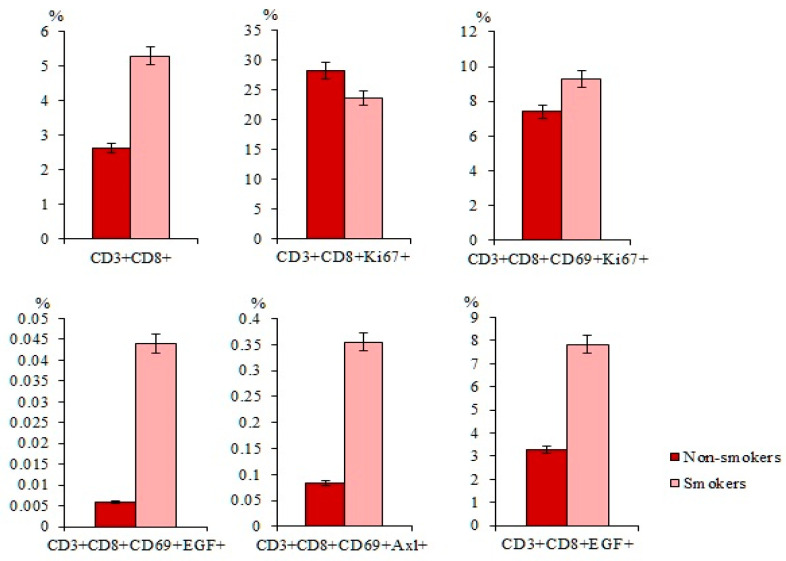
Content of CD3^+^CD8^+^, CD3^+^CD8^+^Ki67^+^, CD3^+^CD8^+^CD69^+^, CD3^+^CD8^+^CD69^+^Ki67^+^, CD3^+^CD8^+^CD69^+^EGF^+^, CD3^+^CD8^+^CD69^+^Axl^+^, and CD3^+^CD8^+^EGF^+^ isolated from the blood of non-smoking and smoking patients with SCLS (percentage of all labelled mononuclear cells). Cells were analyzed by flow cytometry using antibodies against CD3, CD8, CD69, EGF, Axl, and Ki67.

**Figure 7 ijms-23-10853-f007:**
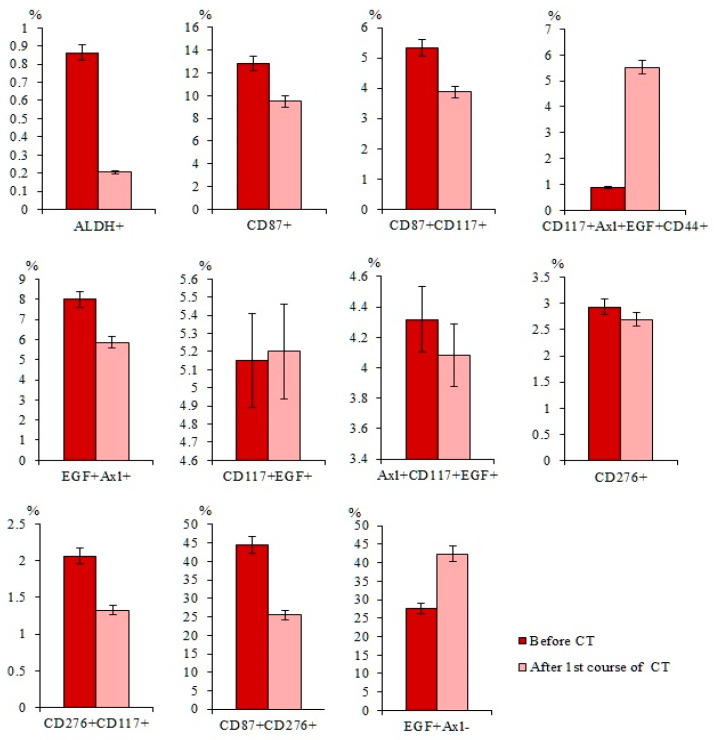
Content of CD87^+^, CD87^+^CD117^+^, CD117^+^Axl^+^EGF^+^CD44^+^, EGF^+^Axl^+^, CD117^+^EGF^+^, Axl^+^CD117^+^EGF^+^, CD276^+^, CD276^+^CD117^+^, CD87^+^CD276^+^, and EGF^+^Axl^−^ circulating tumor cells (CTCs) in the blood of SCLC patients before and after first course of chemotherapy (CT) (% of all labelled mononuclear cells). Cells were analyzed by flow cytometry using antibodies for CD44, CD87, CD117, CD276, EGF, and Axl.

**Figure 8 ijms-23-10853-f008:**
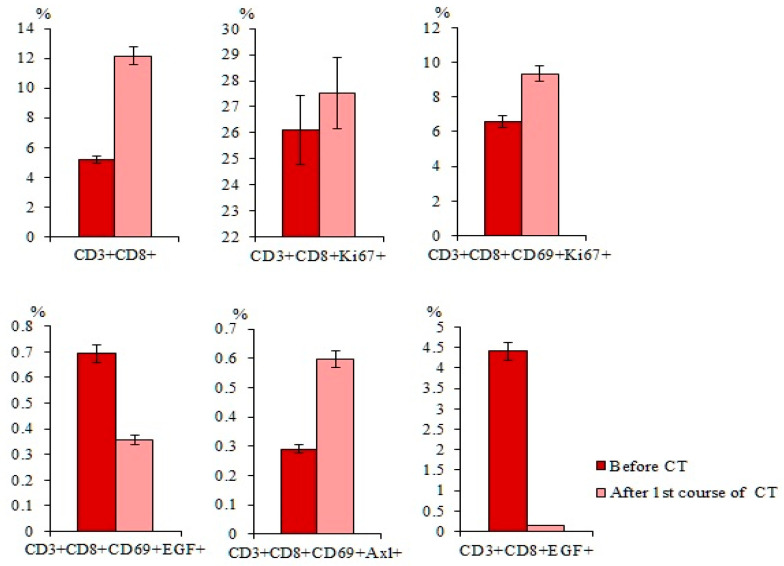
The content of CD3^+^CD8^+^, CD3^+^CD8^+^Ki67^+^, CD3^+^CD8^+^CD69^+^, CD3^+^CD8^+^CD69^+^Ki67^+^, CD3^+^CD8^+^CD69^+^EGF^+^, CD3^+^CD8^+^CD69^+^Axl^+^, and CD3^+^CD8^+^EGF^+^ T-lymphocytes in the blood of SCLC patients before and after first course of chemotherapy (CT) (% of all labelled mononuclear cells). Cells were analyzed by flow cytometry using antibodies against CD3, CD8, CD69, EGF, Axl, and Ki67.

**Figure 9 ijms-23-10853-f009:**
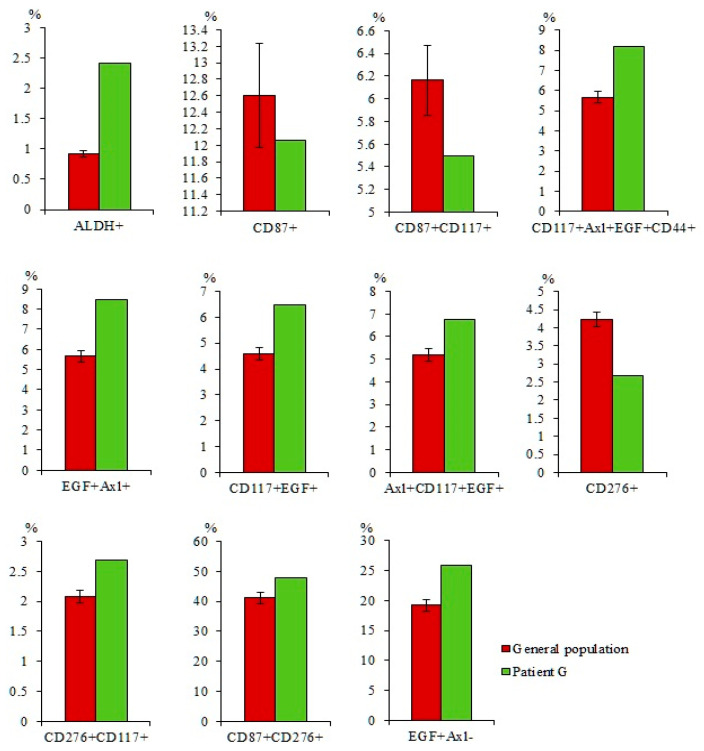
The content of CD87^+^, CD87^+^CD117^+^, CD117^+^Axl^+^EGF^+^CD44^+^, EGF^+^Axl^+^, CD117^+^EGF^+^, Axl^+^CD117^+^EGF^+^, CD276^+^, CD276^+^CD117^+^, CD87^+^CD276^+^, and EGF^+^Axl^−^ circulating tumor cells (CTCs) in the blood of the general population SCLC patients and patient G (% of all labelled mononuclear cells). Cells were analyzed by flow cytometry using antibodies against CD44, CD87, CD117, CD276, EGF, and Axl.

**Figure 10 ijms-23-10853-f010:**
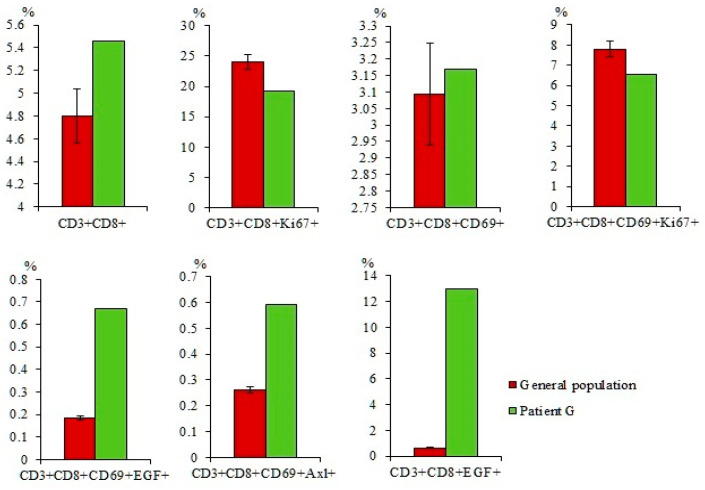
The content of CD3^+^CD8^+^, CD3^+^CD8^+^Ki67^+^, CD3^+^CD8^+^CD69^+^, CD3^+^CD8^+^CD69^+^Ki67^+^, CD3^+^CD8^+^CD69^+^EGF^+^, CD3^+^CD8^+^CD69^+^Axl^+^, and CD3^+^CD8^+^EGF^+^ in the blood of the general population SCLC patients and patient G (% of all labelled mononuclear cells). Cells were analyzed by flow cytometry using antibodies against CD3, CD8, CD69, EGF, Axl, and Ki67.

**Figure 11 ijms-23-10853-f011:**
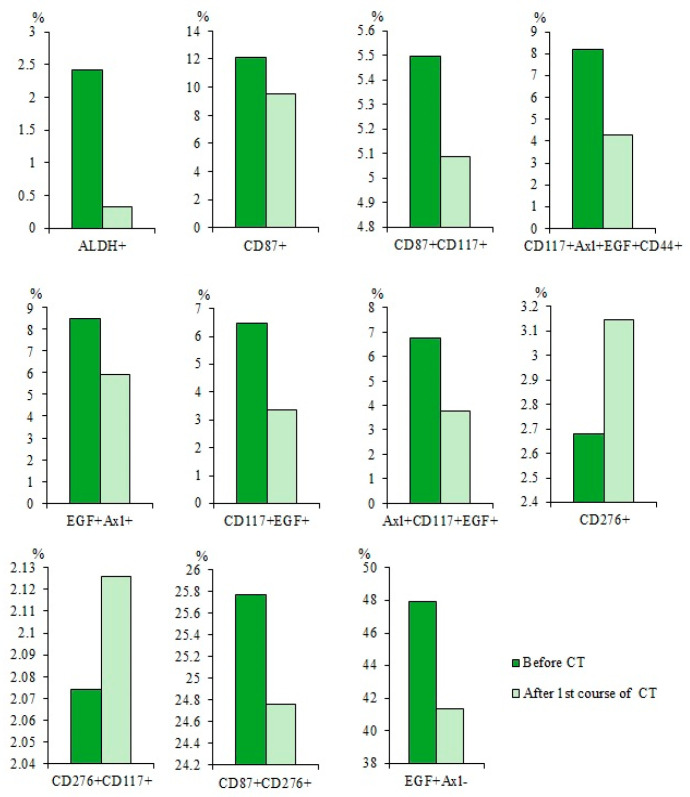
The content of CD87^+^, CD87^+^CD117^+^, CD117^+^Axl^+^EGF^+^CD44^+^, EGF^+^Axl^+^, CD117^+^EGF^+^, Axl^+^CD117^+^EGF^+^, CD276^+^, CD276^+^CD117^+^, CD87^+^CD276^+^, and EGF^+^Axl^−^ CTCs in the blood of SCLC patient G before and after the first course of chemotherapy (CT) (% of all labelled mononuclear cells). Cells were analyzed by flow cytometry using antibodies for CD44, CD87, CD117, CD276, EGF, and Axl.

**Figure 12 ijms-23-10853-f012:**
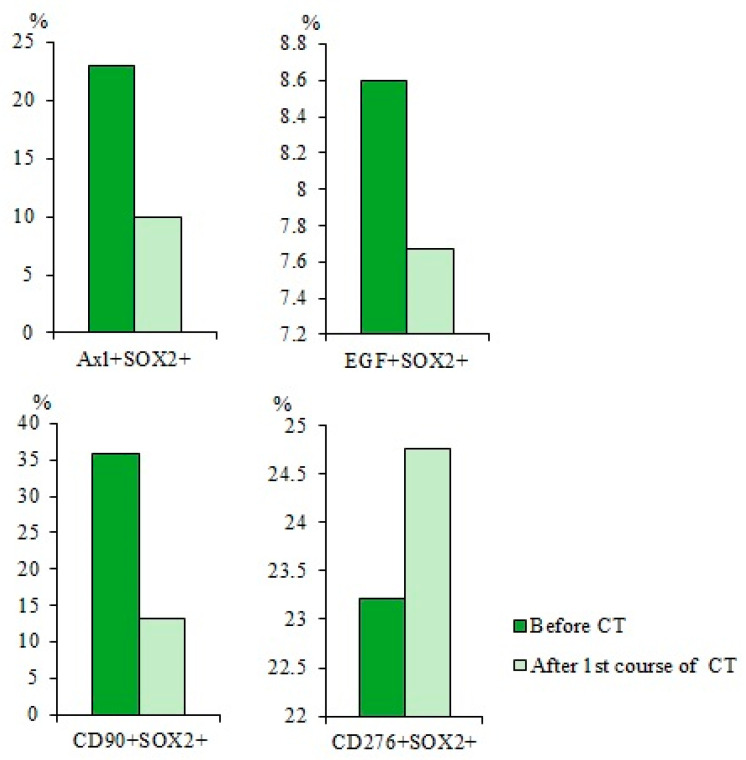
The content of Axl^+^SOX2^+^, EGF^+^SOX2^+^, CD87^+^SOX2^+^, CD276^+^SOX2^+^, and CD90^+^SOX2^+^ circulating tumor cells (CTCs) isolated from the blood of SCLC patients G (% of all labelled mononuclear cells). Cells were analyzed by flow cytometry using antibodies against Axl, EGF, CD87, CD276, CD90, and SOX2. Cells were analyzed by flow cytometry using antibodies for CD87, CD90, CD276, EGF, Axl, and SOX2.

**Figure 13 ijms-23-10853-f013:**
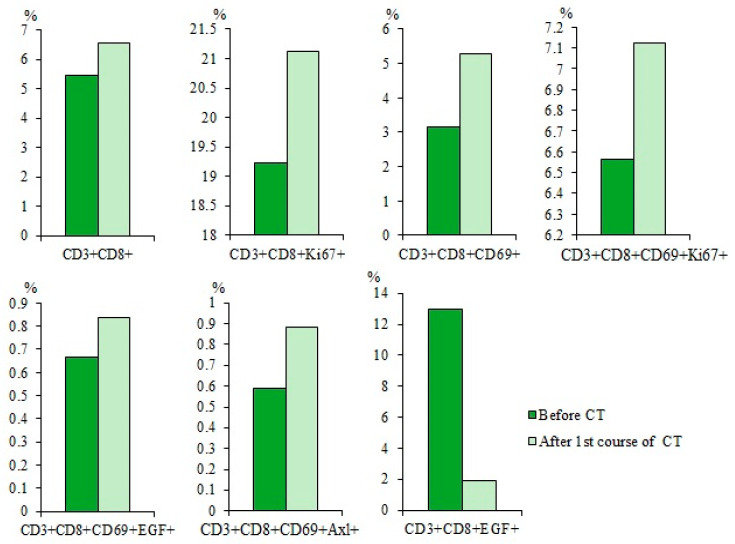
The content of CD3^+^CD8^+^, CD3^+^CD8^+^Ki67^+^, CD3^+^CD8^+^CD69^+^, CD3^+^CD8^+^CD69^+^Ki67^+^, CD3^+^CD8^+^CD69^+^EGF^+^, CD3^+^CD8^+^CD69^+^Axl^+^, and CD3^+^CD8^+^EGF^+^ T-lymphocytes in the blood of SCLC patients G before and after the first course of chemotherapy (CT) (% of all labelled mononuclear cells). Cells were analyzed by flow cytometry using antibodies against CD3, CD8, CD69, EGF, Axl, and Ki67.

**Figure 14 ijms-23-10853-f014:**
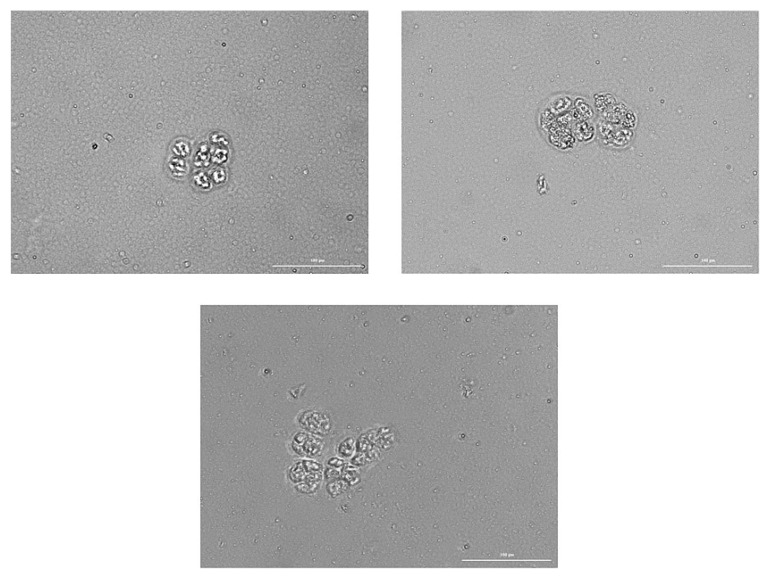
Representative images of spheroids in culture of the adherent fraction of mononuclear cells from patient G after 14 days of culture. Images were obtained using the Cytation 5 Multi-Mode Reader. Native preparations. All scale bars are 100 µm.

**Figure 15 ijms-23-10853-f015:**
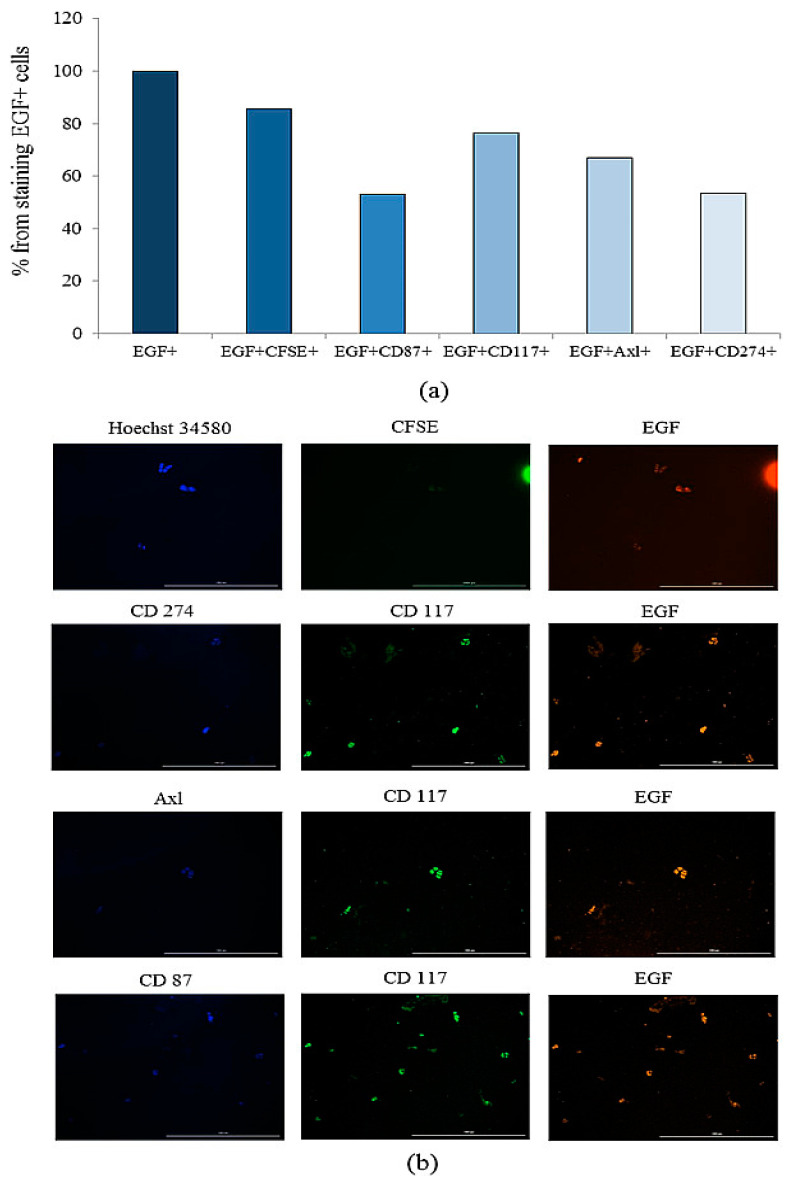
The evaluate of stained single cells in the culture of the adherent mononuclear fraction of patient G. (**a**) The number of cells positive for EGF, EFG/CFSE, EGF/CD87, EGF/CD117, EGF/Axl, and EGF/CD274; (**b**) 20× images of cells stained by: Hoechst 34580 (blue) to identify cell nuclei: CD87 (blue), CD274 (blue), Axl (blue), CFSE (green), CD117 (green), and EGF (yellow). All scale bars are 1000 μm. Cells were analyzed with the Cytation 5 multimodal imager using anti-human CD87 (BV421), CD117 (BB515), CD274 (BV421), Axl (BV421), and EGF (AF647) antibodies and CFSE fluorescent label.

**Figure 16 ijms-23-10853-f016:**
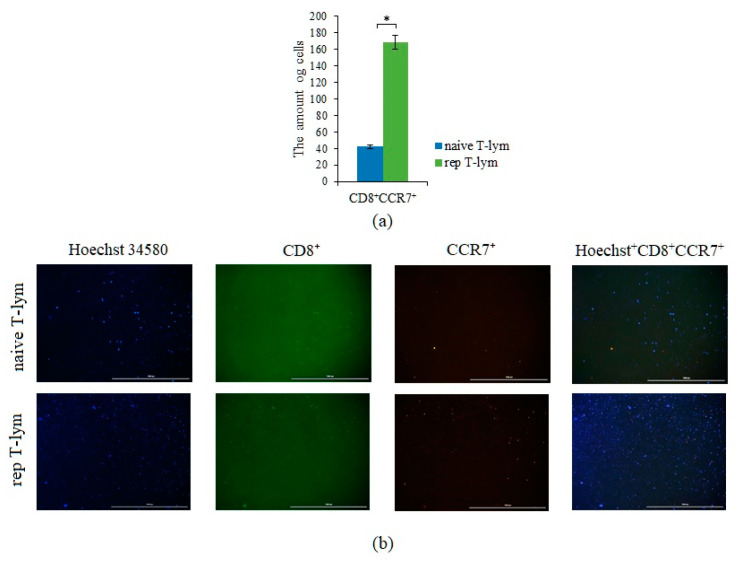
The count of CCR7^+^ T-cells in a culture of naive and reprogrammed CD8^+^ T-lymphocytes isolated from the blood of healthy volunteer. (**a**) The count of naive and reprogrammed CD8+ T-lymphocytes isolated from the blood of healthy volunteer expressing the CCR7 marker in T-lymphocyte culture; (**b**) 4× images of T-cells stained with: Hoechst 34580 (blue) to identify cell nuclei: CD8 FITC (green), CCR7 AF555 (yellow), and (Hoechst^+^CD8^+^CCR7^+^) composite image using all three colors. Determination of the percentage of cells CD8+CCR7+ is made by the ratio of cells counted in green and red channel to total cells counted using blue (DAPI) channel. All scale bars are 1000 µm. *—for comparison with the naive T-lymphocyte by Mann–Whitney test (*p* < 0.05).

**Figure 17 ijms-23-10853-f017:**
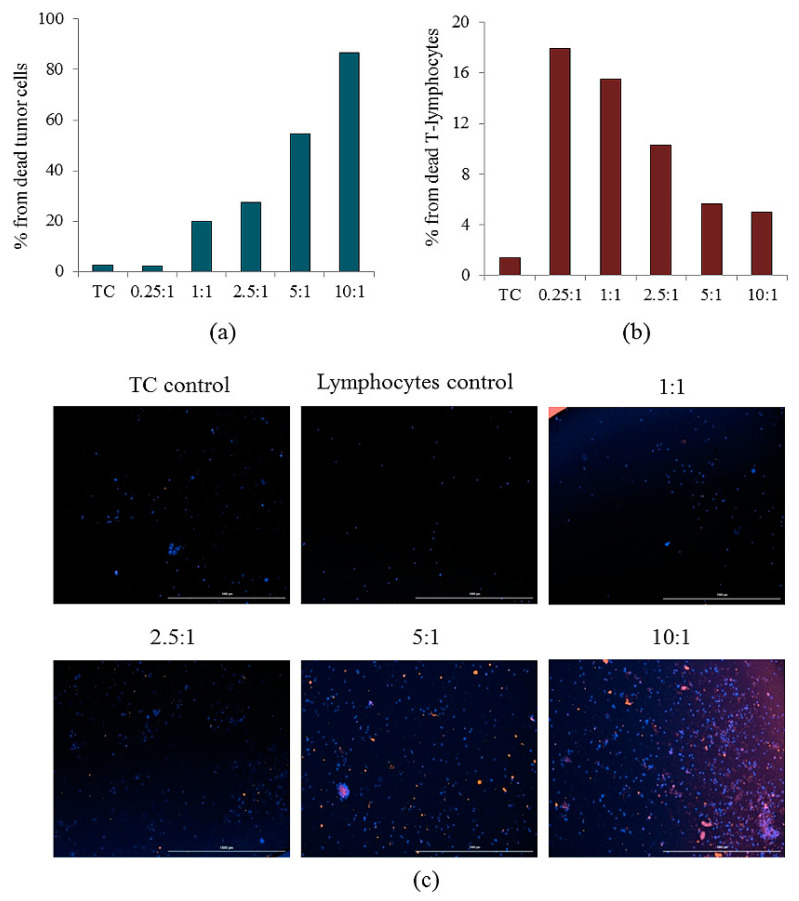
Cytotoxicity and apoptosis of reprogrammed CD8 T-lymphocytes isolated from healthy donor in the primary culture of the adherent mononuclear fraction isolated from the blood of patient G. (**a**) The count of apoptotic tumor cells in the primary culture of the adherent mononuclear fraction isolated from the blood of patient G and (**b**) apoptotic rTcell after cocultivation; (**c**) staining of cell nuclei—Hoechst 34580, T-lymphocytes—CFSE, and dead cells—7-AAD. All scale bars are 1000 µm. Magnification ×4; 1:1, 2.5:1, 5:1, and 10:1 is the ratio of lymphocytes from a healthy donor and culture cells of the adherent mononuclear fraction of patient G. TC—tumor cells of the adherent mononuclear fraction of patient G.

**Figure 18 ijms-23-10853-f018:**
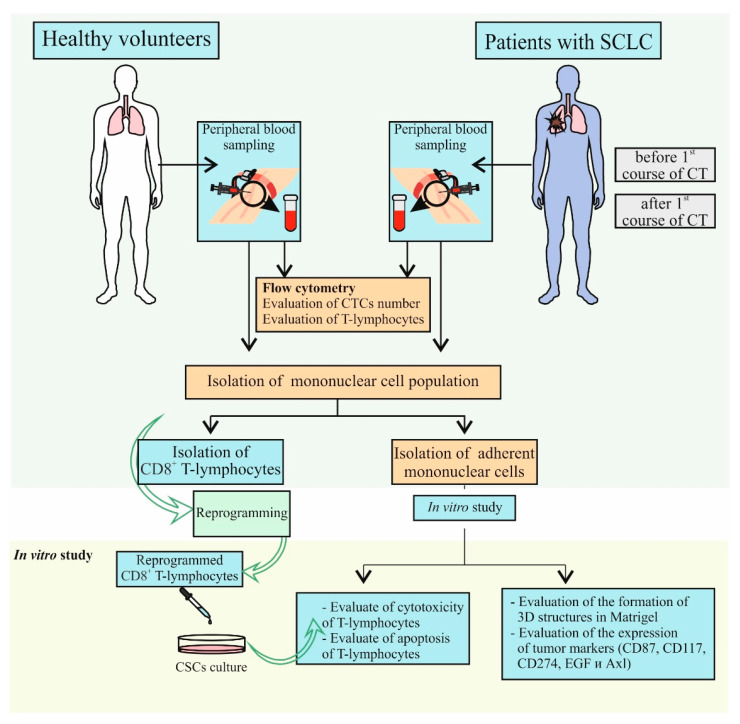
The experimental design. CT—chemotherapy, SCLC—small cell lung cancer.

**Figure 19 ijms-23-10853-f019:**
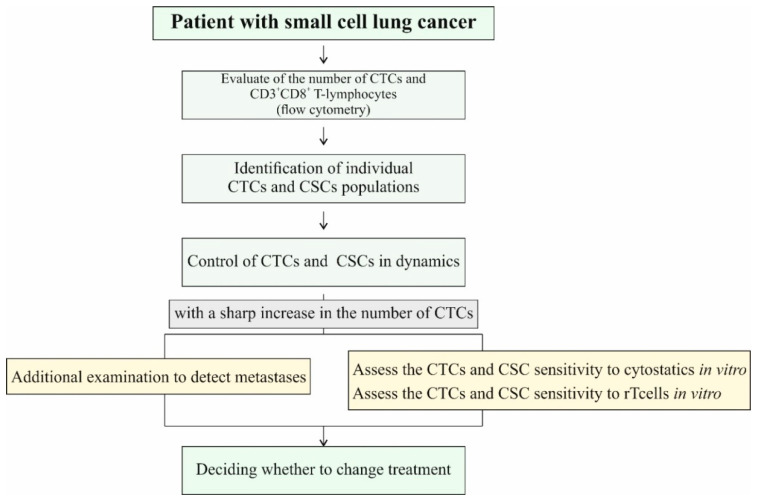
Algorithm for dynamic monitoring of the effectiveness of the treatment.

**Table 1 ijms-23-10853-t001:** Spheroids’ differentiation depending on the number of cells.

Number of Cells	Number of Spheroids	% of the Total Number of Spheroids
from 10 to 14	64	63.37
from 15 to 19	31	30.69
from 20 to 24	3	2.97
from 25 to 29	2	1.98
from 20 to 34	1	0.99

**Table 2 ijms-23-10853-t002:** Cytotoxicity of reprogrammed and naive blood CD3^+^CD8^+^ T-lymphocytes of volunteer K in the primary culture of the adherent mononuclear fraction isolated from the blood of patient G (% dead cells in tumor culture) (M ± m).

CD3^+^CD8^+^ T-Lymphocytes	Ratio of Reprogrammed and Naive CD3^+^CD8^+^ T-Lymphocytes to Tumor Cells (CSCs) (rTcells/CSCs and nTcells/CSCs)					
	0:1 (Tumor Control)	0.25:1	1:1	2.5:1	5:1	10:1
Naive	2.68 ± 0.21	6.07 ± 0.79	15.54 ± 1.53	24.19 ± 0.82	27.24 ± 3.26	31.36 ± 2.2
Reprogrammed	2.68 ± 0.21	5.42 ± 0.61	12.30 ± 2.12	30.21 ± 2.61	46.28 ± 3.59	70.06 ± 2.84

**Table 3 ijms-23-10853-t003:** Spheroids’ differentiation depending on the cell apoptosis and their interaction with reprogrammed CD3^+^CD8^+^ T-lymphocytes in the culture of patient G’s CSCs (% of the total number) (M ± m).

Ratio rTcells/CSCs	Spheroids without Apoptosis and Inclusion rTcells	Spheroids with Apoptosis and Inclusion rTcells	Spheroids with Apoptosis and without Inclusion
0:1 (tumor control)	100	100	100
0.25:1	90.47 ± 1.68	6.06 ± 0.96	3.07 ± 0.97
1:1	76.50 ± 1.66	20.39 ± 2.13	3.11 ± 0.99
2.5:1	61.62 ± 2.68	32.43 ± 2.30	5.98 ± 0.82
5:1	30.98 ± 2.01	58.73 ± 2.18	10.30 ± 0.92
10:1	2.28 ± 1.02	24.32 ± 2.31	73.41 ± 2.24

## Data Availability

Not applicable.

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
