# Peer review of "Analysis of Circulating Tumor and Cancer Stem Cells Provides New Opportunities in Diagnosis and Treatment of Small Cell Lung Cancer"

_ijms, 2022, doi:10.3390/ijms231810853_

Round 1
Reviewer 1 Report
1. The presented paper deeply analyzed Circulating Tumour and Cancer Stem Cells of Small Cell Lung Cancer by flow cytometry and primary culture staining. The issue of Cancer Stem Cells is indeed very significant for cancer research.
2. Lines 41-43 - please, use up-to-date statistics.
3. Figure 3 / Figure 4 / Figure 8: the use of similar staining for "Volunteers - Patients", "Non-smokers - Smokers" and "Before ST - After 1st course of CT" lead to misinterpretation of results. I recommend the different paints for various groups.
4. Figures 9 and 10: What do the standard errors for the patient's G values mean?
5. Along the paper the authors refer to the number of cells, although the figures show the percentage of stained cells.
6. Line 422-424: "Some reports suggest that in the presence of EGFR ligand secreted by tumour cells, CD3+СD8+CD69+Ki67+ T-lymphocytes produce more IFN-γ and TNF-α and show a stronger antitumor response." - this sentence needs the references.
7. Line 490 / Line 711-713: "rTcells were 490 characterized by a high level of CCR7 expression (Figure S1)" - the supplementary figures and tables should be transferred into the main text of the manuscript.
8. The authors need to draw a table with the clinical parameters of patient G and volunteer K.
9. Statistics: Authors should use corrections for multiple comparisons.
I recommend accepting this manuscript after minor revisions.
Author Response
We thank the reviewer for their time and valuable comments. We have now revised the manuscript according to the suggestions. Please find below the reviewers’ comments and our responses. All changes have been included in the revised manuscript.
- The presented paper deeply analyzed Circulating Tumour and Cancer Stem Cells of Small Cell Lung Cancer by flow cytometry and primary culture staining. The issue of Cancer Stem Cells is indeed very significant for cancer research.
Response
- Thank you for positive value of our paper.
- Lines 41-43 - please, use up-to-date statistics.
Response
- We now added the information in our manuscript. We wanted to show dynamics of change the incidence of lung cancer.
- Figure 3 / Figure 4 / Figure 8: the use of similar staining for "Volunteers - Patients", "Non-smokers - Smokers" and "Before ST - After 1st course of CT" lead to misinterpretation of results. I recommend the different paints for various groups.
Response
We now changed the information in our manuscript.
- Figures 9 and 10: What do the standard errors for the patient's G values mean?
Response
We now corrected the information in Figures 9 and 10.
- Along the paper the authors refer to the number of cells, although the figures show the percentage of stained cells.
Response
We now changed the information in our manuscript. We used mononuclear cells isolated from blood for flow cytometric analysis. We excluded the contamination of cell populations with false positive events and the fluorescence associated with non-specific events through the optimization of MoAb panels, proper compensation for the staining with each individual fluorescently conjugated MoAb to maximize signal to noise ratio. Similar to previously published cytometry protocols mononuclear cells from blood were analyzed in logarithmic dot plots, and CSCs, CTCs and T-lymphocytes identification was determined by placement of population gates, which were based on isotype controls. Therefore we refer to the number of cells, although the figures show the percentage of labeled cells.
- Line 422-424: "Some reports suggest that in the presence of EGFR ligand secreted by tumour cells, CD3+СD8+CD69+Ki67+ T-lymphocytes produce more IFN-γ and TNF-α and show a stronger antitumor response." - this sentence needs the references.
Response
We now added the reference in the text.
- Line 490 / Line 711-713: "rTcells were 490 characterized by a high level of CCR7 expression (Figure S1)" - the supplementary figures and tables should be transferred into the main text of the manuscript.
Response
We added the Figure with CCR7 expression (Figure S1) into the main text of the manuscript.
- The authors need to draw a table with the clinical parameters of patient G and volunteer K.
Response
We added the information about clinical parameters of patient G and volunteer K in Table S2.
- Statistics: Authors should use corrections for multiple comparisons.
Response
Thank you for question. Our study was pilot. We know that our study has limitations. We conducted a short-term, single-centre study with a relatively small sample size which reduces the likelihood of generalization. We plan to use corrections for multiple comparisons in further our study when number of patient is increased.
Reviewer 2 Report
I should recommend major revision.
Here are the points:
“Chemotherapy and radiation therapy remain an important component of SCLC treatment [5,6]. However, these SCLC treatments are very effective”. (Line 50-51). There is something wrong in these two following sentences together.
There are many grammatical errors.
The following paragraphs 2 and 3 must be logically connected. There are many repeats and redundancy.
“Kim C.G. et al. showed that a decrease in the content of PD-1+CD8+ T-lymphocytes in the blood of patients with non-small cell lung cancer is associated with a better response to therapy with PD-1 inhibitors”. This study is about non-small cell lung cancer but the paper must include studies with small cell lung cancer.
7 patients with IIС–IV SCLC, aged from 45 to 75 years is too low to have confidential data. The age difference is also high in the range. Please explain this situation.
Aldehyde dehydrogenase is so common in also healthy cells. In this study, is there a specificity? Please explain this situation.
Author Response
We thank the reviewers for their time and valuable comments. We have now revised the manuscript according to the suggestions. All changes have been included in the revised manuscript.
“Chemotherapy and radiation therapy remain an important component of SCLC treatment [5,6]. However, these SCLC treatments are very effective”. (Line 50-51). There is something wrong in these two following sentences together.
Response
Thank you for the question. We now corrected mistake in the text.
There are many grammatical errors.
Response
We now corrected mistakes in the text.
The following paragraphs 2 and 3 must be logically connected. There are many repeats and redundancy.
Response
We now corrected mistakes in the text.
“Kim C.G. et al. showed that a decrease in the content of PD-1+CD8+ T-lymphocytes in the blood of patients with non-small cell lung cancer is associated with a better response to therapy with PD-1 inhibitors”. This study is about non-small cell lung cancer but the paper must include studies with small cell lung cancer.
Response
Thank you for the question. It is known PD-1 expression on CD8 T cells can be related both to their differentiation stage and their activation status. The information about the content of PD-1+CD8+ T-lymphocytes in the blood of patients with SCLC are not enough. However it is known PD-1 expression on CD8 T cells can be related both to their differentiation stage and their activation status. Circulating CD8+ T-lymphocyte can be used as a marker of tumor progression in SCLC [1]. Positive expression of PD-1 or PD-L1 combined with a higher ratio of CD3, CD4, and CD8 was associated with higher relapse-free survival than was negative expression of PD-1 or PD-L1 combined with a lower ratio of CD3, CD4, and CD8 in patients with SCLC [2]. Therefore it is important therapy aimed at increasing the level and activity of CD8+ T-lymphocytes. We used this reference because Kim C.G. et al. showed that a decrease in the content of PD-1+CD8+ T-lymphocytes in the blood and response to therapy with PD-1 inhibitors.
- An N, Wang H, Jia W, Jing W, Liu C, Zhu H, Yu J. The prognostic role of circulating CD8+ T cell proliferation in patients with untreated extensive stage small cell lung cancer. J Transl Med. 2019 Dec 3;17(1):402. doi: 10.1186/s12967-019-02160-7
- Sun C, Zhang L, Zhang W, Liu Y, Chen B, Zhao S, Li W, Wang L, Ye L, Jia K, Wang H, Wu C, He Y, Zhou C. Expression of PD-1 and PD-L1 on Tumor-Infiltrating Lymphocytes Predicts Prognosis in Patients with Small-Cell Lung Cancer. Onco Targets Ther. 2020 Jul 3;13:6475-6483. doi: 10.2147/OTT.S252031
7 patients with IIС–IV SCLC, aged from 45 to 75 years is too low to have confidential data. The age difference is also high in the range. Please explain this situation.
Response
Thank you for question. We agree with you, that 7 patients with IIС–IV SCLC, aged from 45 to 75 years is not enough. However lung cancer is a serious threat to human health. According to the results of WHO statistics from 2001 to 2019, lung cancer ranks among the top ten causes of death in the world every year. Moreover SCLC comprises about 15% of lung cancer while NSCLC comprises approximately 85%. Recruiting patients with SCLC is long time. Our study was pilot. We propose that our approach can be applied to new drugs and prognostic evaluation. We believe that it is already now necessary to publish our approach. We know that our study has limitations. We plan to continue our study to confirm our approach.
Aldehyde dehydrogenase is so common in also healthy cells. In this study, is there a specificity? Please explain this situation.
Response
Thank you for the interesting question. ALDH is an enzyme that catalyzes the oxidation of aldehydes to carboxylic acids to protect cells from oxidative stress. ALDH has been shown to be important for the maintenance of normal HSCs, and it is also commonly used as a marker to differentiate CSCs from different cancers. Previously we showed that ALDH1 expression level in blood of patients with breast cancer can act as personalized diagnostic marker, predictor of complications and the effectiveness of breast cancer treatment. We did not observed a significant increase in the number of ALDH1+ cells circulating in the blood of SCLC patients. Perhaps we have a not enough patients in our study. We know that our study has limitations. We now plan to continue our study to confirm our approach.
Reviewer 3 Report
It seems inappropriate to apply various markers of cancer stem cells and circulating cancer cells isolated from blood collected from a small number of normal people and lung cancer patients to the entire population of lung cancer patients. In this paper, it seems difficult to find the clinical significance of biomarkers detected in the blood of normal people and cancer patients.It seems that the objectivity of the research process can be recognized on the premise of reproducibility for various biomarkers.
Author Response
We thank the reviewer for their time and valuable comments. We have now revised the manuscript according to the suggestions. Please find below the reviewers’ comments and our responses. All changes have been included in the revised manuscript.
It seems inappropriate to apply various markers of cancer stem cells and circulating cancer cells isolated from blood collected from a small number of normal people and lung cancer patients to the entire population of lung cancer patients. In this paper, it seems difficult to find the clinical significance of biomarkers detected in the blood of normal people and cancer patients.It seems that the objectivity of the research process can be recognized on the premise of reproducibility for various biomarkers.
Response
Thank you for question. We analysed literature about various markers of cancer stem cells and circulating cancer cells in SCLC and other cancer. Evidence has shown that cancer stem cells, a sub-population of cells that share many common characteristics with somatic stem cells, contribute to therapeutic failure at the treatment of cancer. It is known that cancer stem cells have been identified in several solid tumors based on the expression of certain cancer stem cells surface markers. Depending on the type of tissue from which they originate, they can express a variety of markers for each type of cancer stem cells or circulating tumour cells. However, the identification and isolation of cancer stem cells or circulating tumour cells continue to be a challenge for therapeutic development. Targeting human surface expression markers with monoclonal antibodies has shown to be a clinically and commercially established therapy. In cancer stem cells, a combination of several markers could be the best approach for specific targeting of cancer stem cells; for example, in breast cancer, the most highly expressed cancer stem cells markers include CD133, CD44, and aldehyde dehydrogenase (ALDH) [1].Cancer stem cells have been identified by surface markers that are common between different cancer types: CD44, CD90, aldehyde dehydrogenase 1 (ALDH1), CD117, Axl, and EGF. We chose these markers for our study.
Our study was pilot. We propose that our approach can be applied to new drugs and prognostic evaluation. We know that our study has limitations. We now plan to continue our study to confirm our approach.
- Croker AK, Goodale D, Chu J, Postenka C, Hedley BD, Hess DA, Allan AL. High aldehyde dehydrogenase and expression of cancer stem cell markers selects for breast cancer cells with enhanced malignant and metastatic ability. J Cell Mol Med. 2009;13(8B):2236–2252. doi: 10.1111/j.1582-4934.2008.00455.x.
Round 2
Reviewer 2 Report
Accept.
Author Response
We thank reviewer for their time and valuable comments. We now improved our manuscript and have revised the manuscript according to the suggestions. All changes have been included in the revised manuscript.
Reviewer 3 Report
1. line 113: need to check symbol
2. Check the figures and legends and unify them
3. line 268: Is there no legend about Figure?
4. Check the description in the table.
5. Carefully check the description of tables and figures in the text.
Author Response
We thank reviewer for their time and valuable comments. We now improved our manuscript and have revised the manuscript according to the suggestions. All changes have been included in the revised manuscript.
Comments and Suggestions:
- line 113: need to check symbol
Response:
Thank you for question. We now corrected symbol.
- Check the figures and legends and unify them
Response:
We now changed the figures and legends and unify them.
- line 268: Is there no legend about Figure?
Response:
There is legend about Figure 11 on line 270. Figure on line 268 is absent.
- Check the description in the table.
Response:
We checked and corrected the description in the table.
- Carefully check the description of tables and figures in the text.
Response:
We now corrected the description of tables and figures in the text.